

# Comparative investigation of parallel spatial interpolation algorithms for building large-scale digital elevation models

Jingzhi Tu, Guoxiang Yang, Pian Qi, Zengyu Ding and Gang Mei

School of Engineering and Technology, China University of Geoscience (Beijing), Beijing, China

## ABSTRACT

The building of large-scale Digital Elevation Models (DEMs) using various interpolation algorithms is one of the key issues in geographic information science. Different choices of interpolation algorithms may trigger significant differences in interpolation accuracy and computational efficiency, and a proper interpolation algorithm needs to be carefully used based on the specific characteristics of the scene of interpolation. In this paper, we comparatively investigate the performance of parallel Radial Basis Function (RBF)-based, Moving Least Square (MLS)-based, and Shepard's interpolation algorithms for building DEMs by evaluating the influence of terrain type, raw data density, and distribution patterns on the interpolation accuracy and computational efficiency. The drawn conclusions may help select a suitable interpolation algorithm in a specific scene to build large-scale DEMs.

## INTRODUCTION

Digital Elevation Model (DEM) is a numerical representation of topography made up of equal-sized grid cells, each with a value of elevation. One of the most important scientific challenges of digital elevation modeling is the inefficiency of most interpolation algorithms in dealing with a large amount of data produced by large-scale DEM with a fine resolution. To solve the problem, one of the common strategies is to parallelize interpolation algorithms on various High Performance Computing (HPC) platforms.

For different large-scale DEM, different parallel spatial interpolation algorithms are usually specifically selected, because a variety of spatial interpolation algorithms exist that behave differently for different data configurations and landscape conditions. Consequently, the accuracy of a DEM is sensitive to the interpolation technique, and it is significant to understand how the various algorithms affect a DEM. Therefore, this study is being conducted.

Spatial interpolation is a category of important algorithms in the field of geographic information. *Siu-Nganlam (1983)* had a review of various interpolation algorithms,

Corresponding authors
Guoxiang Yang, yanggx@cugb.edu.cn
Gang Mei, gang.mei@cugb.edu.cn

including most distance-weighting methods, Kriging, spline interpolation, interpolating polynomials, finite-difference methods, power-series trend models, Fourier models, distance-weighted least-squares, and least-squares fitting with splines. Many spatial interpolation algorithms are used to build DEMs, for example, the Shepard's method (IDW) (*Shepard, 1968*), the Kriging method (*Krige, 1953*), the Discrete Smoothing Interpolation (DSI) method (*Mallet, 1997*), the Radial Basis Function (RBF)-based method (*Powell, 1977*), and the Moving Least Squares (MLS)-based method (*Lancaster & Salkauskas, 1981*).

Much research work (*Gumus & Sen, 2013*; *Chaplot et al., 2006*; *Aguilar et al., 2005*; *Khairnar et al., 2015*; *Polat, Uysal & Toprak, 2015*; *Rishikeshan et al., 2014*) has been conducted to evaluate the effects of different interpolation methods on the precision of DEM interpolation. In the comparative investigation of spatial interpolation algorithms for building DEMs, quite a few studies specifically focused on the impact of data samples and terrain types on interpolation accuracy; among them, *Gumus & Sen (2013)* compared the accuracy of various interpolation methods at different point distributions, the interpolation performance of IDW is worse than other algorithms for the same data distribution. For the same algorithm, in the case of using all points and grid, their experimental results show that the best interpolation performances are Modified Shepard's (MS) for random distribution; Multiquadric Radial Basis Function (MRBF) for curvature distribution, and Inverse Distance Weighted (IDW) for uniform distribution.

*Chaplot et al. (2006)* and *Aguilar et al. (2005)* evaluated the effects of landform types and the density of the original data on the accuracy of DEM production, their results show that interpolation algorithms perform well at higher sampling densities, and MRBF provided significantly better interpolation than IDW in rough or non-uniform terrain. At lower sampling densities, when the spatial structure of height was strong, Kriging yielded better estimates. When the spatial structure of height was weak, IDW and Regularized Spline with Tension (RST) performed better. On the other hand, MRBF performed well in the mountainous areas and Ordinary Kriging (OK) was the best for multi-scales interpolations in the smooth landscape. In addition, *Zhang (2013)* established a descriptive model of local terrain features to study the correlation of surface roughness indicators and spatial distribution indicators for DEM interpolation algorithms. (*Chaplot et al., 2006*). *Ghandehari, Buttenfield & Farmer (2019)* illustrated that the Bi-quadratic and Bi-cubic interpolation methods outperform Weighted Average, Linear, and Bi-linear methods at coarse resolutions and in rough or non-uniform terrain. *Aguilar et al. (2005)* pointed out that MRBF is better than Multilog function for low sample densities and steeper terrain.

With the increasing size of DEMs, it is increasingly necessary to design parallel solutions for existing sequential algorithms to speed up processing. When adopting an interpolation method to deal with a large DEM, the computational cost would be quite expensive, and the computational efficiency would especially be unsatisfied.

The techniques in HPC are widely used to improve computational efficiency in various science and engineering applications such as surface modeling (*Yan et al., 2016*), spatial point pattern analysis (*Zhang, Zhu & Huang, 2017*), urban growth simulation (*Guan et al., 2016*), Delaunay Triangulation (DT) for GIS (*Coll & Guerrieri, 2017*), spatial

interpolation (*Wang, Guan & Wu, 2017*; *Cheng, 2013*; *Mei, 2014*; *Mei, Xu & Xu, 2017*; *Mei, 2014*; *Mei, Xu & Xu, 2016*; *Ding et al., 2018b*), and image processing (*Wasza et al., 2011*; *Lei et al., 2011*; *Yin et al., 2014*; *Wu, Deng & Jeon, 2018*).

One of the effective strategies to solve the problem is to perform the DEM interpolation in parallel on various parallel computing platforms such as shared-memory computers, distributed-memory computers, or even clusters. The parallelization of DEM interpolation can be developed with the computational power of modern multicore Central Processing Units (CPUs) and many-core Graphics Processing Units (GPUs). For example, *Zhou et al. (2017)* proposed a parallel Open Multi-Processing (OpenMP)- and Message Passing Interface (MPI)-based implementation of the Priority-Flood algorithm that identifies and fills depressions in raster DEMs. *Yan et al. (2015)* accelerated high-accuracy surface modeling (HASM) in constructing large-scale and fine resolution DEM surfaces by the use of GPUs and applied this acceleration algorithm to simulations of both ideal Gaussian synthetic surfaces and real topographic surfaces in the loess plateau of Gansu province. *Tan et al. (2017)* presented a novel method to generate contour lines from grid DEM data, based on the programmable GPU pipeline, that can be easily integrated into a 3D GIS system. *Chen et al. (2010)* demonstrated a new algorithm for reconstructing contour maps from raster DEM data for digital-earth and other terrain platforms in real-time entirely based on modern GPUs and programmable pipelines.

The RBF, Kriging, MLS and Shepard's interpolation algorithms are the most frequently used spatial interpolation algorithms, among which, the Kriging method can be regarded as an instance of RBF framework (*Peng et al., 2019*). Therefore, in this paper, we comparatively investigate the performance of the RBF-based, MLS-based, and Shepard's interpolation algorithms for building DEMs by evaluating the influence of terrain type, raw data density, and distribution patterns on the interpolation accuracy and computational efficiency.

The rest of the paper is organized as follows. 'Background' briefly introduces the basic principles of eight interpolation methods. 'Methods' concentrates mainly on our parallel implementations of the eight interpolation methods and creation of the testing data. 'Results' introduces some of the experimental tests performed on the CPU and GPU. 'Discussion' discusses the experimental results. Finally, 'Conclusion' states conclusions from the work.

# BACKGROUND

In this section, we briefly introduce eight spatial interpolation algorithms.

## MLS-based Interpolation Algorithms

The MLS method obtains the fitting surface by solving the equation group derived from minimizing the sum of the squares of the errors between the fitting data and the given node data.

### *Original MLS Interpolation Algorithm*

The MLS approximation is used to approximate field variables and their derivatives. In a domain $\Omega$, the MLS approximation $f^h(x)$ of the field variable $f(x)$ in the vicinity of a

point $\bar{x}$ is given as

$$f^h(x) = \sum_{j=1}^{m} p_j(x) \cdot a_j(\bar{x}) = P^T(x) \cdot a(\bar{x}) \tag{1}$$

where $p_j(x), j = 1, 2, \ldots, m$ is a complete basis function with coefficients $a_j(\bar{x})$. At each point $\bar{x}$, $a_j(\bar{x})$ is chosen to minimize the weighted residual $L_2-$ norm ($L_2-$ norm refers to $\|x\|_2$, where $x = [x_1, x_2, \ldots, x_n]^T$, and $\|x\|_2 = \sqrt{(|x_1|^2 + |x_2|^2 + |x_3|^2 + \cdots + |x_n|^2)}$):

$$J = \sum_{I=1}^{N} w(\bar{x} - x_I) \left[ P^T(x_I) a(\bar{x}) - f_I \right]^2 \tag{2}$$

where $N$ is the number of nodes in the compact-supported neighborhood of $\bar{x}$ and $f_I$ refers to the nodal parameter of $f$ at $x = x_I$. Nodes refer to data points in the compact-supported neighborhood of $\bar{x}$. Compact-supported, i.e., point $\bar{x}$ is only related to the nodes of its neighborhood, $x_I$ is one of the nodes in the compact-supported neighborhood. And $w(x - x_k)$ is the compact-supported weight function. The most commonly used weight functions are the spline functions, for example, the cubic spline weight function (Eq. (3)):

$$w(\bar{s}) = \begin{cases} \dfrac{2}{3} - 4\bar{s}^2 + 4\bar{s}^3, & \bar{s} \leq \dfrac{1}{2} \\ \dfrac{4}{3} - 4\bar{s} + 4\bar{s}^2 - \dfrac{4}{3}\bar{s}^3, & \dfrac{1}{2} < \bar{s} \leq 1 \\ 0, & \bar{s} > 1 \end{cases} \tag{3}$$

where $\bar{s} = \frac{s}{s_{\max}}$ and $s = \bar{x} - x_I$.

The minimum of $J$ with respect to $a(\bar{x})$ gives the standard form of MLS approximation:

$$f^h(x) = \sum_{I=1}^{N} \phi_I(x) f_I = \Phi(x) F. \tag{4}$$

### Orthogonal MLS interpolation algorithm

For a given polynomial basis function $p_i(x)$, $i = 1, 2, \cdots, m$, there is an orthonormal basis function $q_i(x, \bar{x})$ that satisfies:

$$q_1(x, \bar{x}) = p_1(x)$$

$$q_i(x, \bar{x}) = p_i(x) - \sum_{j=1}^{i-1} \alpha_{ij}(x, \bar{x}) q_j(x, \bar{x}), i = 2, 3, \cdots, m \tag{5}$$

where $\alpha_{ij}(x, \bar{x})$ is the coefficient that makes $q_i(x, \bar{x})$ perpendicular to $q_j(x, \bar{x})$.

$$\alpha_{ij}(\bar{x}) = \frac{\sum_{k=1}^{N} w_k(\bar{x}) p_i(x_k) q_j(x_k, \bar{x})}{\sum_{k=1}^{N} w_k(\bar{x}) q_j^2(x_k, \bar{x})} \tag{6}$$

Because the coefficient matrix is a diagonal matrix, the solution for $a_i(x)$ does not require matrix inversion, i.e.,

$$a_i(\bar{x}) = \frac{\sum_{k=1}^{N} w_k(\bar{x}) q_i(x_k, \bar{x}) f_k}{\sum_{k=1}^{N} w_k(\bar{x}) q_i^2(x_k, \bar{x})} \tag{7}$$

where $a_i$ and $a_j(\bar{x})$ (Eq. (1)) have the same definition. $f_k$ and $f_I$ (Eq. (2)) have the same definition, i.e., the nodal parameter of $f$ at $x = x_k$. Finally, $a_i$ and the orthonormal basis function $q_i(x, \bar{x})$ are fitted into Eq. (1) to obtain the orthogonal MLS approximation $f^h(x)$.

When the number or order of basis functions increases, only $a_{m+1}$ and $\alpha_{m+1}$ need to be calculated in Gram–Schmidt orthogonalization (*Steve, 2011*); recalculation of all entries in the coefficient matrix is not needed. This could reduce the computational cost and the computational error.

### Lancaster's MLS interpolation algorithm

A singular weight function is adopted to make the approximation function $f^h(x)$ constructed by the interpolation type MLS method satisfy the properties of the Kronecker $\delta$ function:

$$\omega(x, x_k) = \begin{cases} \|(x - x_k)/\rho_k\|^{-\alpha}, & \|x - x_k\| \leq \rho_k \\ 0, & \|x - x_k\| > \rho_k \end{cases} \tag{8}$$

Let $p_0(x) \equiv 1, p_1(x), \ldots, p_{\bar{m}}(x)$ denote the basis function used to construct the approximation function, where the number of basis functions is $\bar{m} + 1$. To implement the interpolation properties, a new set of basis functions is constructed for a given basis function. First, $p_0(x)$ are standardized, i.e.,

$$\tilde{p}_0(x, \bar{x}) = \frac{1}{\left[ \sum_{k=1}^{N} \omega(x, x_k) \right]^{1/2}} \tag{9}$$

Then, we construct a new basis function of the following form:

$$\tilde{p}_i(x, \bar{x}) = p_i(\bar{x}) - \sum_{k=1}^{N} \frac{\omega(x, x_k)}{\sum_{l=1}^{N} \omega(x, x_l)} P_i(x_k), i = 1, 2, \ldots, \bar{m}. \tag{10}$$

### RBF-based interpolation algorithm

The RBF operates as a spline, essentially fitting a series of piecewise surfaces to approximate a complex terrain.

Let $X = \{x_1, x_2, \ldots, x_N\}$ be a set of pairwise distinct points in a domain $\Omega \subseteq R^d$ with associated data values $f_i$, $i = 1, 2, \ldots, N$. We consider the problem of construction a $d$-variety function $F \in C^k(R^d)$ that interpolates the known data. Specifically, we require $F(x_i) = f_i, i = 1, 2, \ldots, N$. If we take $F$ in the form.

$$F(x) = \sum_{j=1}^{N} w_j \varphi \left( \|x_i - x_j\|_2 \right) \tag{11}$$

where $\varphi : [0, \infty] \to R$ is a suitable continuous function, the interpolation conditions become:

$$\sum_{j=1}^{N} w_j \varphi \left( \|x_i - x_j\|_2 \right) = f_i, \quad i = 1, 2, \ldots, N. \tag{12}$$

## Shepard's interpolation algorithms

*Shepard (1968)* proposed a series of interpolation algorithms on the basis of weighting averages. These algorithms are termed Shepard's method. The essential idea behind Shepard's method is to estimate expected values of the interpolation point by weighting averages of the nearby discrete points as follows:

Let $(x_i, y_i)$, $i = 1, 2, \ldots, N$ be the interpolation point and $f_i$ be the corresponding value at interpolation point $(x_i, y_i)$. The expected value $f$ at any point can be expressed as

$$f(x) = \sum_{i=1}^{N} \frac{w_i(x) f_i}{\sum_{j=1}^{N} w_j(x)} \tag{13}$$

where $w(x)$ is a weight function.

The differences between the different variants of Shepard's method are in the selection of different weighting functions. In this subsection, four common variants of Shepard's method will be briefly introduced (Eqs. (14)–(19)).

### Variant A of Shepard's interpolation algorithm

First, select the influence radius $R > 0$ and let the weight function be

$$w(r) = \begin{cases} \dfrac{1}{r}, & 0 < r \leq \dfrac{R}{3} \\ \dfrac{27}{4} \left( \dfrac{r}{R} - 1 \right)^2, & \dfrac{R}{3} < r \leq R \\ 0, & r > R \end{cases} \tag{14}$$

Then, a variation of Shepard's interpolation will be obtained.

### Variant B of Shepard's Interpolation Algorithm

When employing the following weight function (Eq. (15)), a new variation of Shepard's interpolation will be obtained.

$$w(\bar{s}) = \begin{cases} \dfrac{2}{3} - 4\bar{s}^2 + 4\bar{s}^3, & \bar{s} \leq \dfrac{1}{2} \\ \dfrac{4}{3} - 4\bar{s} + 4\bar{s}^2 - \dfrac{4}{3}\bar{s}^3, & \dfrac{1}{2} < \bar{s} \leq 1 \\ 0, & \bar{s} > 1 \end{cases} \tag{15}$$

### Inverse Distance Weighted (IDW) interpolation algorithm

If the weight function is selected as

$$w_i(x) = \frac{1}{d(x, x_i)^\alpha} \tag{16}$$

the IDW interpolation is obtained. Typically, $\alpha = 2$ in the standard IDW. Where $d(x, x_i)$ is the distance between the interpolation point $x_i$ and the nearby discrete point $x$.

### AIDW interpolation algorithm

The Adaptive Inverse Distance Weighted (AIDW) is an improved version of the standard IDW (*Shepard, 1968*) originated by *Lu & Wong (2008)*. The distance-decay parameter $\alpha$ is no longer a prespecified constant value but is adaptively adjusted for a specific unknown interpolated point according to the distribution of the nearest neighboring data points.

The parameter $\alpha$ is taken as

$$\alpha(\mu_R) = \begin{cases} \alpha_1, & 0.0 \le \mu_R \le 0.1 \\ \alpha_1[1-5(\mu_R-0.1)]+5\alpha_2(\mu_R-0.1), & 0.1 \le \mu_R \le 0.3 \\ 5\alpha_3(\mu_R-0.3)+\alpha_2[1-5(\mu_R-0.3)], & 0.3 \le \mu_R \le 0.5 \\ \alpha_3[1-5(\mu_R-0.5)]+5\alpha_4(\mu_R-0.5), & 0.5 \le \mu_R \le 0.7 \\ 5\alpha_5(\mu_R-0.7)+\alpha_4[1-5(\mu_R-0.7)], & 0.7 \le \mu_R \le 0.9 \\ \alpha_5, & 0.9 \le \mu_R \le 1.0 \end{cases} \tag{17}$$

$$\mu_R = \begin{cases} 0, & R(S_0) \le R_{\min} \\ 0.5-0.5\cos[\pi(R(S_0)-R_{\min})/R_{\max}], & R_{\min} \le R(S_0) \le R_{\max} \\ 1, & R(S_0) \ge R_{\max} \end{cases} \tag{18}$$

where the $\alpha_1, \alpha_2, \alpha_3, \alpha_4, \alpha_5$ are the to-be-assigned five levels or categories of distance decay value. $R_{\min}$ or $R_{\max}$ refer to a local nearest neighbor statistic value, and $R_{\min}$ and $R_{\max}$ can generally be set to 0.0 and 2.0, respectively. Then,

$$R(S_0) = \frac{2\sqrt{N/A}}{k}\sum_{i=1}^{k}d_i \tag{19}$$

where $N$ is the number of points in the study area, $A$ is the area of the study region, $k$ is the number of nearest neighbor points, $d_i$ is the nearest neighbor distances and $S_0$ is the location of an interpolated point.

## METHODS

### Implementations of the spatial interpolation algorithms

We have implemented the spatial interpolation algorithms of RBF (*Ding et al., 2018b*), MLS (*Ding et al., 2018a*), IDW (*Mei, 2014*), and AIDW (*Mei, Xu & Xu, 2017*) in our previous work. To evaluate the computational performance of the GPU-accelerated interpolation, we implement and compare (1) the sequential implementation, (2) the parallel implementation developed on a multicore CPU, (3) the parallel implementation using a single GPU, and (4) the parallel implementation using multiple GPUs.

There are two key ideas behind the presented spatial interpolation algorithm:

(1) We use an efficient $k$-Nearest Neighbor ($k$NN) search algorithm (*Mei, Xu & Xu, 2016*) to find the local set of data points for each interpolated point.

(2) We employ the local set of data points to compute the prediction value of the interpolated point using different interpolation methods.

*Mei & Tian (2016)* evaluated the impact of different data layouts on the computational efficiency of the GPU-accelerated IDW interpolation algorithm. They implemented three
IDW versions of GPU implementations, based upon five data layouts, including the Structure of Arrays (SoA), the Array of Structures (AoS), the Array of aligned Structures (AoaS), the Structure of Arrays of aligned Structures (SoAoS), and a hybrid layout, then they carried out several groups of experiments to evaluate the impact of different data layouts on the interpolation efficiency. Based on their experimental results, the layout SoA is shown in Listing 1.

```
struct Pt {
    float x[N];
    float y[N];
    float z[N];
};
struct Pt myPts;
```

Listing 1: The layout SoA

The $k$NN (*Cover & Hart, 1967*) is a machine learning algorithm often used in classification, the $k$-Nearest Neighbor means that each data point can be represented by its $k$ nearest neighbor points. In all of the presented interpolation algorithms, for each interpolation point, a local set of data points is found by employing the $k$NN search procedure and the found local sets of data points are then used to calculate the prediction value of the interpolation point. For large size of DEM, the $k$NN search algorithm can effectively improve the speed of interpolation by searching only the points near the interpolation points (*Mei, Xu & Xu, 2016*).

Assuming there are m interpolated points and n data points, the process of the $k$NN search algorithm is as follows:

Step 1: the $k$ distances between the $k$ data points and each of the interpolated points are calculated; for example, if the k is set to 5, then there are 5 distances needed to be calculated; see the row (A) in Fig. 1.

Step 2: The $k$ distances are sorted in ascending order; see the row (B) in Fig. 1.

Step 3: For each of the rest (m-$k$) data points,

(1) The distance $d$ is calculated, for example, the distance is 4.2 ($d = 4.2$);

(2) The $d$ with the $k$th distance are compared: if $d <$ the $k$th distance, then replace the $k$th distance with the $d$ (see row (C));

(3) Iteratively compare and swap the neighboring two distances from the $k$th distance to the 1st distance until all the $k$ distances are newly sorted in ascending order; see the rows (C)–(E) in Fig. 1.

## Creation of the testing data

Two sets of DEM data were downloaded from the Geospatial Data Cloud (http://www.gscloud.cn//). More specifically, two 30-m resolution DEMs for two 20 km × 20 km regions in Hebei and Sichuan provinces were selected. The topography of Hebei province is mainly plain, while the topography of Sichuan province is mainly mountainous. Two sets of DEM data are derived from remote sensing satellites and compiled by the CNIC

| Original | 0.3 | 8.6 | 1.5 | 5 | 6.2 | (A) |
|----------|-----|-----|-----|-----|-----|-----|
| Sorted   | 0.3 | 1.5 | 5   | 6.2 | 8.6 | (B) |
| Replaced | 0.3 | 1.5 | 5   | 6.2 | 4.2 | (C) |
| Swapped  | 0.3 | 1.5 | 5   | 4.2 | 6.2 | (D) |
| Desired  | 0.3 | 1.5 | 4.2 | 5   | 6.2 | (E) |

**Figure 1  An illustration of the process of the $k$NN search algorithm.**

(Computer Network Information Center, Chinese Academy of Sciences). More details on the selected DEMs are presented in Fig. 2.

Data points and interpolated points (listed in Tables 1 and 2) are produced as follows:

(1) The selected DEMs is imported into the software ArcGIS.

(2) A square region $S$ is delimited in selected DEMs. For example, the two 20 km $\times$ 20 km regions shown in Fig. 2.

(3) Generating the $x$ and $y$ coordinates of randomly determined points by random number generation algorithms in the square region $S$, and then accessing the corresponding $z$ coordinates from the DEM (the randomly determined points are the data points $P1$). Evenly distributed (regularly distributed) data points are randomly extracted using the Linear Congruential Random Number Method (Lehmer, 1949), and normally distribution (irregularly distributed, mathematical expectation $\mu = 10,000$, standard deviation $\sigma = 3,333$) data points are randomly extracted using the Box–Muller Method (Box & Muller, 1958). For example, we set Size 1, the extracted regularly distributed data points $P1 = 249,990$ (Table 1), and density is $P1/S_0$ ($S_0$ is the area of $S$, and $S_0$ is a fixed value, where $S_0 = 20$ km $\times$ 20 km).

(4) The square region $S$ is triangulated into a planar triangular mesh using the Del auney algorithm (Watson, 1981), the mesh nodes are considered to be the interpolation points, with known $x$ and $y$ coordinates and unknown $z$ coordinates, the unknown $z$ coordinates is the estimated value to be obtained by interpolation. According to the randomly sampled points obtained in Step 3, we use the interpolation method mentioned in 'Background' to interpolate. Then, the corresponding exact elevation of the interpolation point is obtained by accessing the $z$ value of the DEM at the associated $x$ and $y$ coordinates. Finally, the $z$ values at the mesh points are used as control for testing the accuracy of the interpolated $z$ values.

To quantitatively determine regular and irregular point sampling, Average Nearest Neighbor analysis (Ebdon, 1985) is applied. In the proposed method, Nearest Neighbor Ratio (NNR) is used to evaluate the distribution pattern of sample points: if the NNR > 1, the distribution pattern shows clustered; if the NNR < 1, the distribution pattern shows

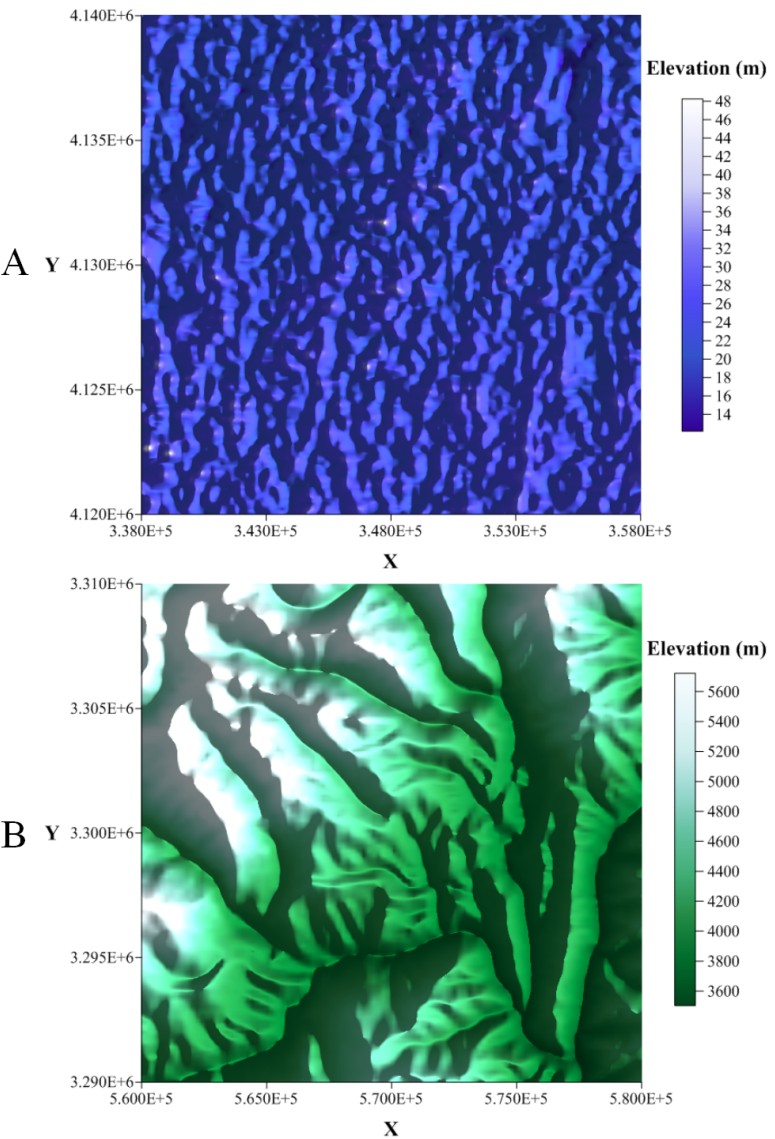

**Figure 2 The selected Zone 1 and Zone 2.** (A) 2.5D model of the Zone 1 study area and (B) 2.5D model of the Zone 2 study area.

dispersed. As listed in Table 3, the NNR of regularly-distributed, approximately 1.001, is greater than 1, the distribution pattern is dispersed (Fig. 3A), that is regularly-distributed; the NNR of irregularly-distributed, approximately 0.78, is less than 1, the distribution pattern is clustered (Fig. 3B), that is irregularly-distributed.

### Zone 1 (Flat Zone)

The first selected region is located in Hengshui City, Hebei Province. The DEM of this region has the identifier ASTGTM_N37E115 and is derived from the Geospatial Data Cloud (http://www.gscloud.cn/). The location and elevation of this region is illustrated in Fig. 2. In the region, the highest elevation is 48 m and the lowest is 8 m. We translated the

**Table 1 Ten used groups of experimental testing data in the Flat zone.**

| Data set | | Number of data points | Number of interpolated points |
|---|---|---|---|
| Regularly-distributed | Size 1 | 249,990 | 259,496 |
| | Size 2 | 499,975 | 529,080 |
| | Size 3 | 999,883 | 1,036,780 |
| | Size 4 | 1,499,750 | 1,540,373 |
| | Size 5 | 1,999,566 | 2,000,520 |
| Irregularly-distributed | Size 1 | 249,920 | 259,496 |
| | Size 2 | 499,751 | 529,080 |
| | Size 3 | 998,840 | 1,036,780 |
| | Size 4 | 1,497,397 | 1,540,373 |
| | Size 5 | 1,995,531 | 2,000,520 |

**Table 2 Ten used groups of experimental testing data in the Rugged zone.**

| Data set | | Number of data points | Number of interpolated points |
|---|---|---|---|
| Regularly-distributed | Size 1 | 249,994 | 259,496 |
| | Size 2 | 499,970 | 529,080 |
| | Size 3 | 999,884 | 1,036,780 |
| | Size 4 | 1,499,746 | 1,540,373 |
| | Size 5 | 1,999,544 | 2,000,520 |
| Irregularly-distributed | Size 1 | 249,924 | 259,496 |
| | Size 2 | 499,728 | 529,080 |
| | Size 3 | 998,867 | 1,036,780 |
| | Size 4 | 1,497,444 | 154,0373 |
| | Size 5 | 1,995,443 | 2,000,520 |

**Table 3 The NNR of regular and irregular point sampling.**

| Data set | | Flat zone | Rugged zone |
|---|---|---|---|
| Regularly-distributed | Size 1 | 1.001731 | 1.001170 |
| | Size 2 | 1.001219 | 1.001291 |
| | Size 3 | 1.001437 | 1.001173 |
| | Size 4 | 1.001987 | 1.001758 |
| | Size 5 | 1.002431 | 1.001869 |
| Irregularly-distributed | Size 1 | 0.783242 | 0.781741 |
| | Size 2 | 0.782947 | 0.784534 |
| | Size 3 | 0.783653 | 0.784086 |
| | Size 4 | 0.784653 | 0.784056 |
| | Size 5 | 0.783745 | 0.784888 |

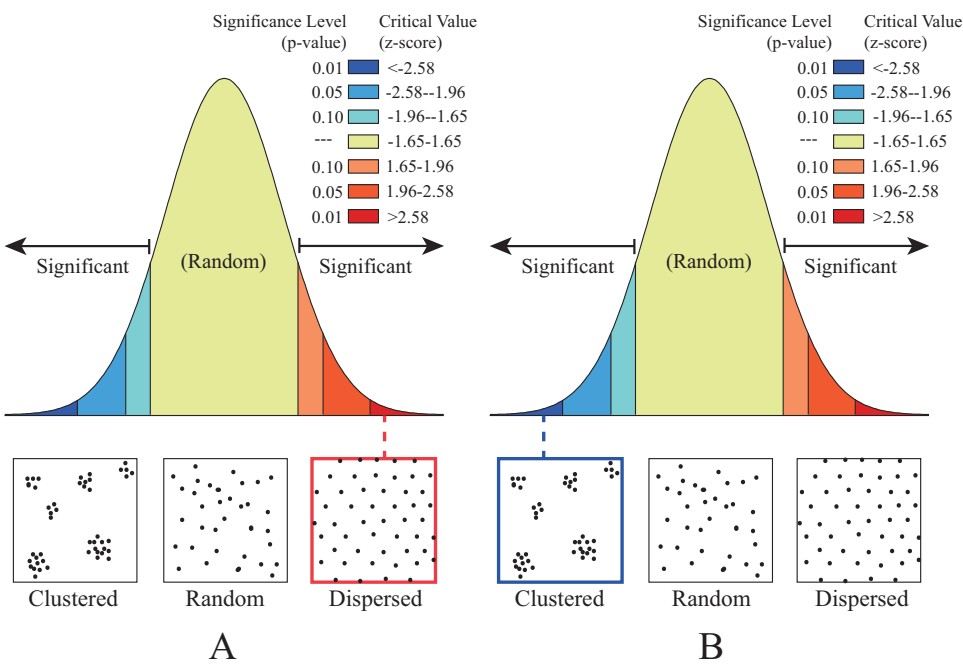

**Figure 3** **The distribution patterns determined by the Average Nearest Neighbor analysis.** (A) Regularly distributed and (B) irregularly distributed.

X coordinate by 348,000 and the Y coordinate by 4,130,000 to obtain a 20 km ×20 km square area centered on the origin. Five sets of benchmark test data were generated in this region; see Table 1.

### Zone 2 (Rugged Zone)
The second selected region is located in Ganzi Tibetan Autonomous Prefecture, Sichuan Province. The DEM of this region has the identifier ASTGTM_N29E099 and is derived from the Geospatial Data Cloud (http://www.gscloud.cn/). The location and elevation of this region is illustrated in Fig. 2. In the region, the highest elevation is 5,722 m and the lowest is 3,498 m. We translated the X coordinate by 570,000 and the Y coordinate by 3,300,000 to obtain a 20 km ×20 km square area centered on the origin. Five sets of benchmark test data are generated in this region; see Table 2.

## Criteria for comparison
In this paper, we evaluate the interpolation algorithms described in 'Background' by: (1) comparing the interpolation accuracy and efficiency when the terrain is gentle and rugged, and (2) comparing the interpolation accuracy and efficiency when data points are evenly distributed and nonuniformly distributed.

The accuracy of each interpolation method is analyzed by comparing the elevation values predicted by the interpolation algorithms with the real DEM elevation value. The efficiency of each interpolation method is compared by benchmarking the running time of

**Table 4** Specifications of the workstation and the software used for the experimental tests.

| Specifications | Details |
| --- | --- |
| CPU | Intel Xeon E5-2650 v3 |
| CPU Frequency | 2.30 GHz |
| CPU RAM | 144 GB |
| CPU Core | 40 |
| GPU | Quadro M5000 |
| GPU Memory | 8 GB |
| GPU Core | 2048 |
| OS | Windows 7 Professional |
| Compiler | Visual Studio 2010 |
| CUDA Version | v8.0 |

different implementations developed in sequence, on a multicore CPU, on a single GPU, and on multiple GPUs.

## RESULTS

### Experimental environment

To evaluate the computational performance of the presented various parallel interpolations, we conducted ten groups of experimental tests in both the flat zone and the rugged zone on a powerful workstation equipped with two Quadro M5000 GPUs. The specifications of the workstations are listed in Table 4.

### Test results of interpolation accuracy for different interpolation algorithms

In this paper, we adopt the Normalized Root-Mean-Square-Error (NRMSE) as the metric to measure the interpolation accuracy of the different interpolation algorithms. The NRMSE is defined in Eq. (20).

Normalized Root-Mean-Square-Error (NRMSE):

$$NRMSE = \frac{1}{\max_{1 \leq i \leq N_i} |f_a|} \sqrt{\frac{1}{N_i} \sum_{i=1}^{N_i} |f_n - f_a|^2} \tag{20}$$

where $N_i$ is the number of interpolated points, $f_a$ is the theoretically exact solution of the $i$th interpolated point (the elevation of the DEM at this point), and $f_n$ is the predicted value of the $i$th interpolated point.

The interpolation accuracy of the ten groups of experimental tests is listed in Table 5. The numerical value shown in Table 5 is NRMSE, which means that the smaller the numerical value, the higher the interpolation accuracy.

As listed in Table 5, the most accurate interpolation algorithm is the MLS interpolation algorithm. For the small size (Size 1), compared with other two algorithms, the MLS algorithm is 13.1%–49.4% more accurate than the RBF algorithm, and it is 2.1%–75.8% more accurate than the Shepard's algorithm. On the other hand, for the same algorithm, when the distribution pattern is the same, its accuracy in the flat area is higher than that

**Table 5 Interpolation accuracy of the parallel interpolation algorithms implemented on a single GPU.**

| | Data set | | Original MLS | Orthogonal MLS | Lancaster's MLS | *k*NN RBF | *k*NN AIDW | *k*NN IDW | *k*NN shepard1 | *k*NN shepard2 |
|---|---|---|---|---|---|---|---|---|---|---|
| Flat zone | Regularly distributed | Size 1 | 7.49E−5 | 7.49E−5 | 7.50E−5 | 9.23E−5 | 1.06E−4 | 1.07E−4 | 1.05E−4 | 1.03E−4 |
| | | Size 2 | 6.25E−5 | 6.25E−5 | 6.03E−5 | 6.85E−5 | 7.92E−5 | 7.98E−5 | 7.81E−5 | 7.80E−5 |
| | | Size 3 | 5.52E−5 | 5.52E−5 | 5.23E−5 | 5.67E−5 | 6.17E−5 | 6.19E−5 | 6.15E−5 | 6.23E−5 |
| | | Size 4 | 5.16E−5 | 5.16E−5 | 4.88E−5 | 5.24E−5 | 5.45E−5 | 5.46E−5 | 5.47E−5 | 5.58E−5 |
| | | Size 5 | 4.91E−5 | 4.91E−5 | 4.64E−5 | 4.99E−5 | 5.05E−5 | 5.05E−5 | 5.08E−5 | 5.20E−5 |
| | Irregularly-distributed | Size 1 | 1.96E−4 | 1.96E−4 | 1.86E−4 | 2.14E−4 | 1.90E−4 | 1.95E−4 | 1.98E−4 | 2.02E−4 |
| | | Size 2 | 1.53E−4 | 1.53E−4 | 1.48E−4 | 1.71E−4 | 1.57E−4 | 1.60E−4 | 1.62E−4 | 1.65E−4 |
| | | Size 3 | 1.20E−4 | 1.20E−4 | 1.15E−4 | 1.36E−4 | 1.28E−4 | 1.31E−4 | 1.32E−4 | 1.33E−4 |
| | | Size 4 | 1.07E−4 | 1.07E−4 | 1.02E−4 | 1.21E−4 | 1.15E−4 | 1.17E−4 | 1.18E−4 | 1.19E−4 |
| | | Size 5 | 9.50E−5 | 9.50E−5 | 9.14E−5 | 1.07E−4 | 1.05E−4 | 1.05E−4 | 1.06E−4 | 1.07E−4 |
| Rugged zone | Regularly-distributed | Size 1 | 2.23E−4 | 2.23E−4 | 2.58E−4 | 4.41E−4 | 9.21E−4 | 9.26E−4 | 9.43E−4 | 9.69E−4 |
| | | Size 2 | 1.23E−4 | 1.23E−4 | 1.35E−4 | 2.35E−4 | 6.13E−4 | 6.16E−4 | 6.35E−4 | 6.63E−4 |
| | | Size 3 | 9.09E−5 | 9.09E−5 | 9.07E−5 | 1.37E−4 | 4.13E−4 | 4.12E−4 | 4.33E−4 | 4.58E−4 |
| | | Size 4 | 8.13E−5 | 8.13E−5 | 7.99E−5 | 1.08E−4 | 3.31E−4 | 3.30E−4 | 3.50E−4 | 3.71E−4 |
| | | Size 5 | 7.62E−5 | 7.62E−5 | 7.48E−5 | 9.39E−5 | 2.85E−4 | 2.83E−4 | 3.02E−4 | 3.21E−4 |
| | Irregularly-distributed | Size 1 | 3.37E−3 | 3.37E−3 | 3.02E−3 | 3.99E−3 | 4.06E−3 | 4.12E−3 | 4.11E−3 | 4.07E−3 |
| | | Size 2 | 1.98E−3 | 1.98E−3 | 1.88E−3 | 2.96E−3 | 3.49E−3 | 3.55E−3 | 3.57E−3 | 3.52E−3 |
| | | Size 3 | 1.03E−3 | 1.03E−3 | 1.10E−3 | 1.56E−3 | 2.02E−3 | 2.05E−3 | 2.04E−3 | 2.02E−3 |
| | | Size 4 | 8.15E−4 | 8.15E−4 | 8.21E−4 | 1.16E−3 | 1.70E−3 | 1.70E−3 | 1.68E−3 | 1.67E−3 |
| | | Size 5 | 6.33E−4 | 6.33E−4 | 6.59E−4 | 9.78E−4 | 1.35E−3 | 1.36E−3 | 1.36E−3 | 1.37E−3 |

the rugged area. For example, for the MLS algorithm, when the distribution pattern is nonuniformly distributed, the accuracy of the Lancaster' MLS algorithm in the flat area is approximately 90% higher than that of the Lancaster' MLS algorithm in the rugged area.

As shown in Figs. 4 and 5, the NRMSEs of various interpolation methods for the regularly distributed are less than 50% of the NRMSEs of various interpolation methods for the irregularly distributed. The above behavior becomes even more obvious in the rugged zone than in the flat zone. Thus, the regular distribution provides a more accurate solution for both the rugged and the flat areas.

## Test results of computational efficiency for different interpolation algorithms

In our experimental tests, the value of *k* is 20. Those twenty groups of experimental tests were performed on the workstations mentioned above. The running times and corresponding speedups of each group of experimental tests are presented in the following section. The speedup is defined in Eq. (21).

$$speedup = \frac{T_{seq}}{T_{par}} \tag{21}$$

where $T_{seq}$ is the running time of sequential implementation, and $T_{par}$ is the running time of parallel implementation.

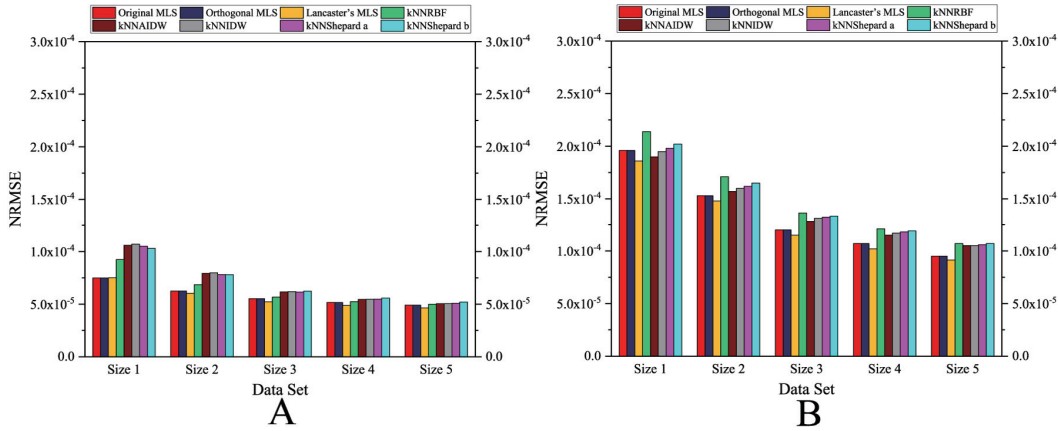

**Figure 4 Interpolation accuracy of GPU-accelerated interpolation algorithms in the Flat zone.** (A) Regularly distributed and (B) irregularly distributed.

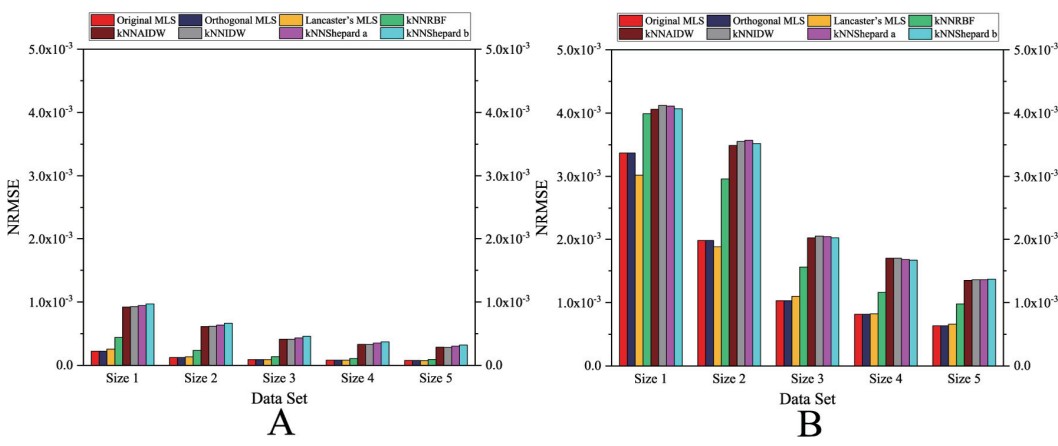

**Figure 5 Interpolation accuracy of GPU-accelerated interpolation algorithms in the Rugged zone.** (A) Regularly distributed and (B) irregularly distributed.

### *Computational efficiency of sequential implementations*

As listed in Table 6, for the sequential version, when giving the same sets of data points and interpolation points, the order of computational time from fastest to slowest is: the Shepard's interpolation method, the MLS interpolation, and the RBF interpolation. The computational time of Shepard's interpolation method is approximately 20% less than the MLS interpolation method, and it is approximately 70% less than the computational time of the computational time of RBF interpolation method.

### *Computational efficiency of parallel implementations*

As shown in Figs. 6– 11, the parallel version developed on multi-GPUs has the highest speedup in the three parallel versions. Except for the RBF interpolation method, the maximum speedups of other interpolation algorithms are greater than 45.

**Table 6  Running time (ms) of sequential implementations.**

|  | Data set | Original MLS | Orthogonal MLS | Lancaster's MLS | *k*NN RBF | *k*NN AIDW | *k*NN IDW | *k*NN shepard1 | *k*NN shepard2 |
|---|---|---|---|---|---|---|---|---|---|
| Flat zone | Regularly-distributed | | | | | | | | |
| | Size 1 | 1,571.33 | 1,501.67 | 1,613.00 | 4,194.33 | 1,520.67 | 1,239.00 | 1,290.67 | 1,270.33 |
| | Size 2 | 3,253.33 | 3,238.33 | 3,330.33 | 8,547.33 | 3,100.67 | 2,475.67 | 2,618.33 | 2,583.00 |
| | Size 3 | 6,355.67 | 6,063.33 | 6,487.67 | 16,610.67 | 6,154.67 | 4,957.33 | 5,196.33 | 5,125.33 |
| | Size 4 | 9,462.00 | 9,036.67 | 9,670.33 | 24,856.67 | 9,161.33 | 7,359.00 | 7,754.67 | 7,674.00 |
| | Size 5 | 12,403.33 | 11,854.00 | 12,725.33 | 32,370.33 | 12,050.67 | 9,643.33 | 10,230.67 | 10,058.00 |
| | Irregularly-distributed | | | | | | | | |
| | Size 1 | 1,458.33 | 1,392.00 | 1,500.00 | 4,028.67 | 1,409.00 | 1,104.33 | 1,177.33 | 1,157.67 |
| | Size 2 | 3,042.33 | 2,919.67 | 3,115.00 | 8,291.33 | 2,923.00 | 2,300.33 | 2,430.67 | 2,397.33 |
| | Size 3 | 6,067.00 | 5,738.00 | 6,129.00 | 16,299.33 | 5,783.67 | 4,559.00 | 4,834.67 | 4,776.33 |
| | Size 4 | 8,856.00 | 8,491.33 | 9,142.00 | 24,286.00 | 8,636.33 | 6,779.33 | 7,211.33 | 7,105.00 |
| | Size 5 | 11,706.00 | 11,214.00 | 12,031.33 | 31,744.00 | 11,354.00 | 8,922.00 | 9,498.00 | 9,372.67 |
| Rugged zone | Regularly-distributed | | | | | | | | |
| | Size 1 | 1,576.00 | 1,497.67 | 1,605.33 | 4,148.00 | 1,512.67 | 1,204.67 | 1,278.00 | 1,264.00 |
| | Size 2 | 3,211.33 | 3,131.00 | 3,285.33 | 8,452.33 | 3,117.33 | 2,620.33 | 2,695.33 | 2,582.67 |
| | Size 3 | 6,354.33 | 6,064.67 | 6,500.33 | 16,649.33 | 6,139.67 | 4,898.00 | 5,200.33 | 5,127.67 |
| | Size 4 | 9,444.67 | 9,026.67 | 9,662.33 | 24,811.67 | 9,187.00 | 7,293.33 | 7,710.33 | 7,660.33 |
| | Size 5 | 12,416.67 | 11,853.33 | 12,711.33 | 32,372.67 | 12,008.33 | 9,606.33 | 10,205.67 | 10,062.00 |
| | Irregularly-distributed | | | | | | | | |
| | Size 1 | 1,503.00 | 1,408.00 | 1,516.00 | 4,060.33 | 1,424.00 | 1,117.33 | 1,191.67 | 1,214.67 |
| | Size 2 | 3,032.33 | 2,883.33 | 3,110.33 | 8,274.33 | 2,925.67 | 2,277.00 | 2,424.00 | 2,391.33 |
| | Size 3 | 5,943.33 | 5,704.67 | 6,089.33 | 16,226.67 | 5,746.33 | 4,534.00 | 4,800.33 | 4,735.67 |
| | Size 4 | 8,920.00 | 8,524.33 | 9,132.33 | 24,262.00 | 8,654.67 | 6,781.67 | 7,224.00 | 7,115.67 |
| | Size 5 | 11,632.33 | 11,147.33 | 11,925.33 | 31,612.00 | 11,282.33 | 8,885.33 | 9,435.67 | 9,320.33 |

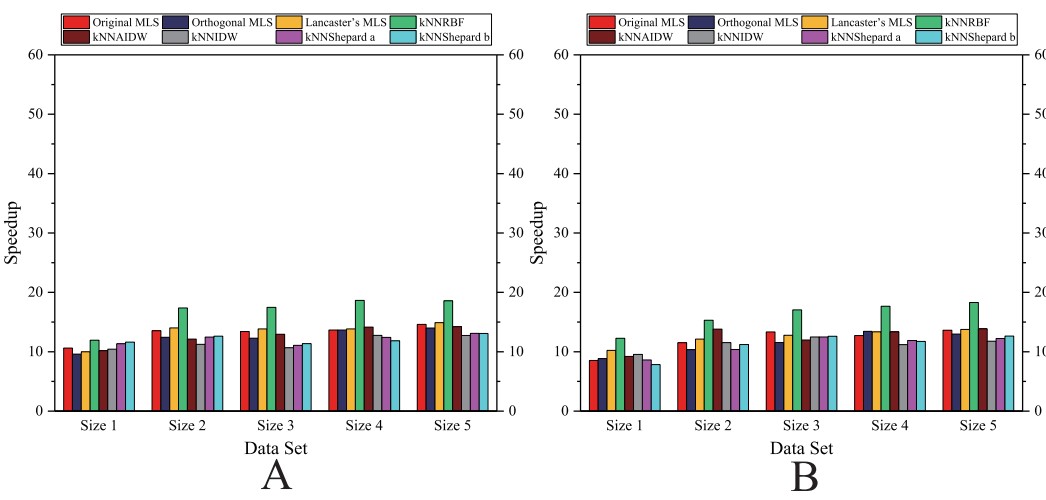

**Figure 6  Comparison of the speedups of the parallel implementations developed on a multicore CPU in the Flat zone.** (A) Regularly distributed and (B) irregularly distributed.

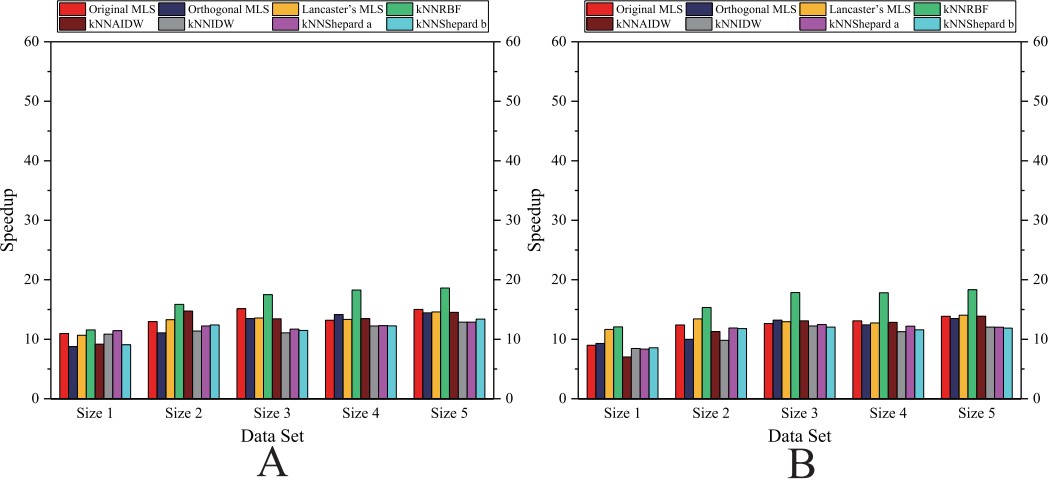

**Figure 7** **Comparison of the speedups of the parallel implementations developed on a multicore CPU in the Rugged zone.** (A) Regularly distributed and (B) irregularly distributed.

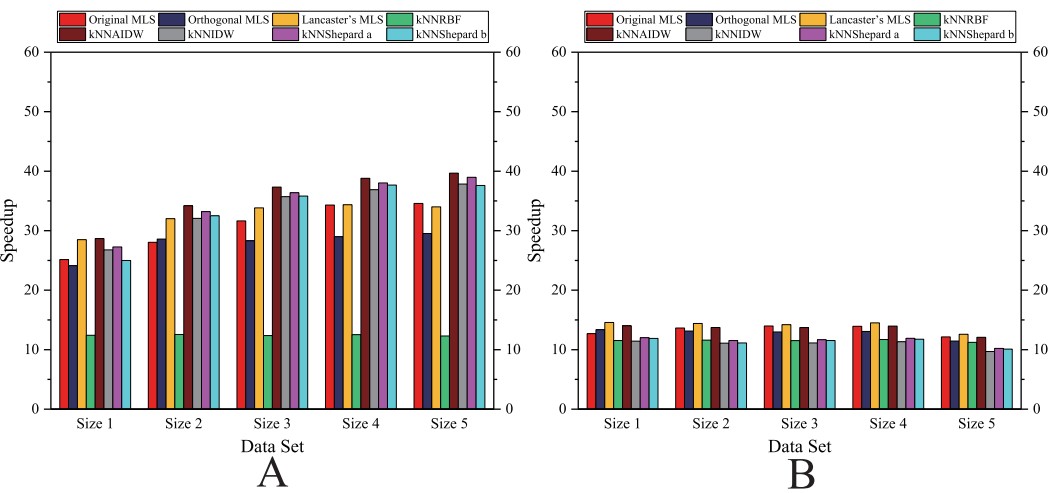

**Figure 8** **Comparison of the speedups of the parallel implementations developed on a single GPU in the Flat zone.** (A) Regularly distributed and (B) irregularly distributed.

As shown in Figs. 12 and 13, for the parallel version developed on multi-GPUs, the order of the computational time from fastest to slowest is: the Shepard's interpolation, the MLS interpolation, the RBF interpolation method. The computational time of Shepard's interpolation method is 3%–30% less than the computational time of the MLS interpolation method, and it is 70%–85% less than the computational time of the RBF interpolation method.

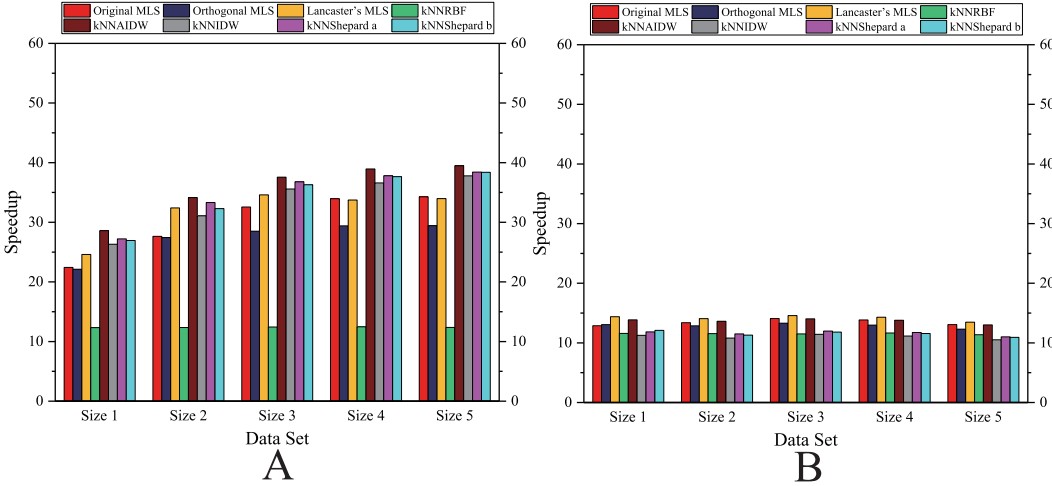

**Figure 9  Comparison of the speedups of the parallel implementations developed on a single GPU in the Rugged zone.** (A) Regularly distributed and (B) irregularly distributed.

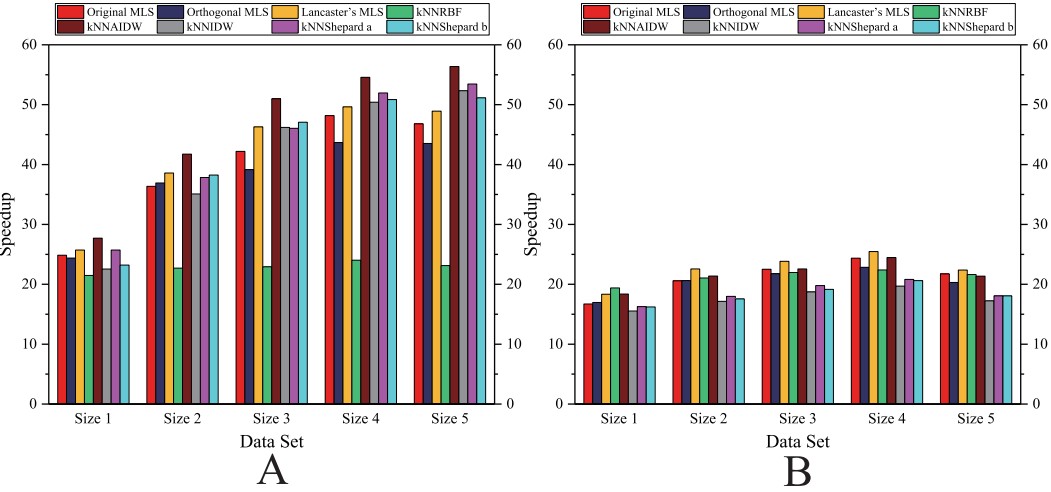

**Figure 10  Comparison of the speedups of the parallel implementations developed on multi-GPUs in the Flat zone.** (A) Regularly distributed and (B) irregularly distributed.

## DISCUSSION

The interpolation accuracy and computational efficiency are two critical issues that should be considered first in any interpolation algorithms. The interpolation accuracy should first be satisfied; otherwise, numerical analysis results would be inaccurate. In addition, the computational efficiency should be practical.

More specifically, in the subsequent section we will analyze (1) the interpolation accuracy of the presented eight GPU-accelerated interpolation algorithms with different data sets and (2) the computational efficiency of the presented eight interpolation algorithms.

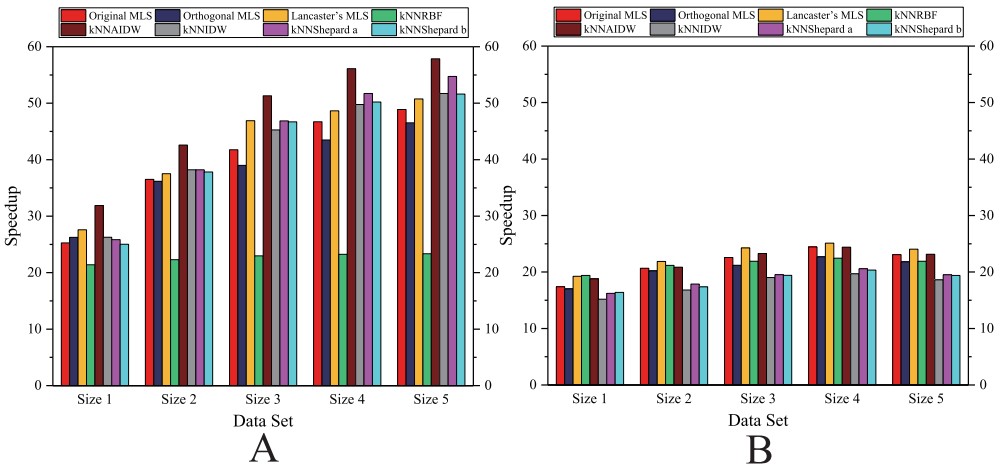

**Figure 11** **Comparison of the speedups of the parallel implementations developed on multi-GPUs in the Rugged zone.** (A) Regularly distributed and (B) irregularly distributed.

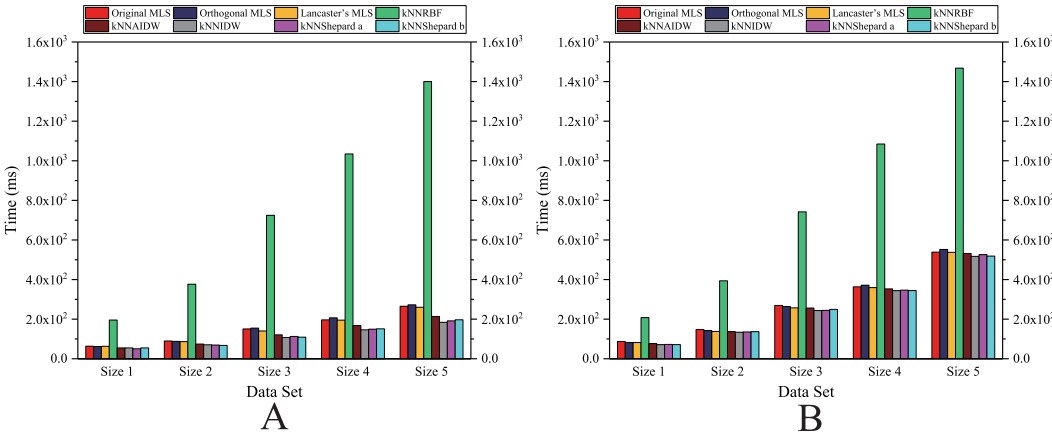

**Figure 12** **Comparison of the running time of the parallel implementations developed on multi-GPUs in the Flat zone.** (A) Regularly distributed and (B) irregularly distributed.

## Comparison of interpolation accuracy

To better compare the accuracy of the described interpolation algorithms, in the case of the highest sample density (Size 5) and the lowest sample density (Size 1), we listed those algorithms with the highest accuracy (i.e., the minimum NRMSE) in Table 7.

As listed in Table 7, for lower sample density (Size 1), the Original MLS algorithm has the best interpolation performance in regularly distributed. However, for higher sample density (Size 5), in general, the improved MLS algorithm Lancaster's MLS has higher interpolation accuracy than the Original MLS. In particular, the Original MLS has best accuracy in the rugged zone with irregularly distributed interpolation points.

On the other hand, for Shepard's interpolation algorithms, the $k$NNAIDW is an improved version of the IDW, which can adaptively determine the power parameter

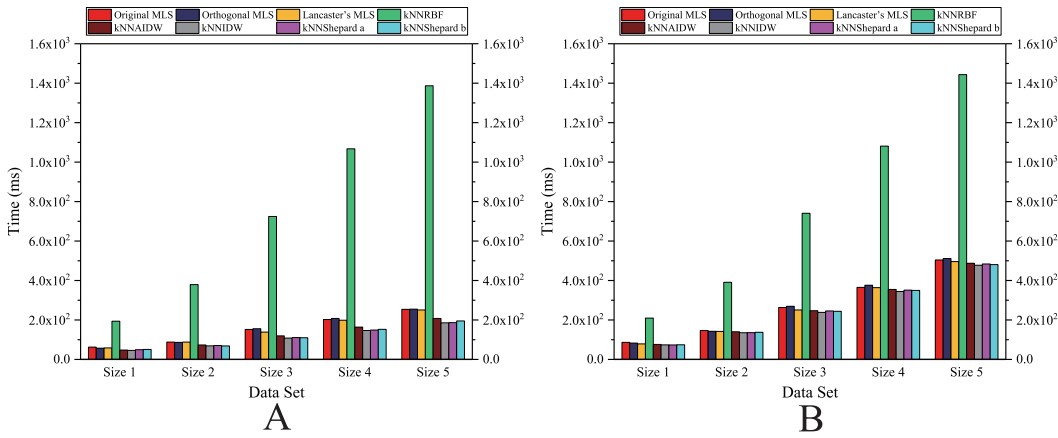

**Figure 13** **Comparison of the running time of the parallel implementations developed on multi-GPUs in the Rugged zone.** (A) Regularly distributed and (B) irregularly distributed.

**Table 7** **The algorithm with the highest accuracy in congeneric algorithms and its corresponding NRMSE.**

| Data set | | | MLS algorithm | RBF algorithm | Shepard's interpolation algorithm |
|---|---|---|---|---|---|
| Flat zone | Regularly-distributed | Size 1 | Original MLS (7.49E–5) | kNNRBF (9.23E–5) | kNNShepard2 (1.03E–4) |
| | | Size 5 | Lancaster's MLS (4.64E–5) | kNNRBF (4.99E–5) | kNNAIDW (5.05E–5) |
| | Irregularly distributed | Size 1 | Lancaster's MLS (1.86E–4) | kNNRBF (2.14E–4) | kNNAIDW (1.90E–4) |
| | | Size 5 | Lancaster's MLS (9.14E–5) | kNNRBF (1.07E–4) | kNNAIDW (1.05E–4) |
| Rugged zone | Regularly-distributed | Size 1 | Original MLS (2.23E–4) | kNNRBF (4.41E–4) | kNNAIDW (9.21E–4) |
| | | Size 5 | Lancaster's MLS (7.48E–5) | kNNRBF (9.39E–5) | kNNIDW (2.83E–4) |
| | Irregularly-distributed | Size 1 | Lancaster's MLS (3.02E–3) | kNNRBF (3.99E–3) | kNNAIDW (4.06E–3) |
| | | Size 5 | Original MLS (6.33E–4) | kNNRBF (9.78E–4) | kNNAIDW (1.35E–3) |

according to the spatial points' distribution pattern. Therefore, in Shepard's interpolation algorithms, the $k$NNAIDW has higher accuracy in most situations. Although under some specific conditions, the $k$NNShepard2 and $k$NNIDW have higher accuracy than $k$NNAIDW, the accuracy of $k$NNAIDW is quite similar to them.

As listed Table 7. For the same flat zone, when the data points are uniformly distributed, the order of the interpolation accuracy from high to low is: the MLS interpolation algorithm, RBF, and Shepard's interpolation method; when the data points are normal distribution, the order of the interpolation accuracy from high to low is: the MLS interpolation algorithm, Shepard's interpolation method, and RBF. For the same rugged zone, regardless of the

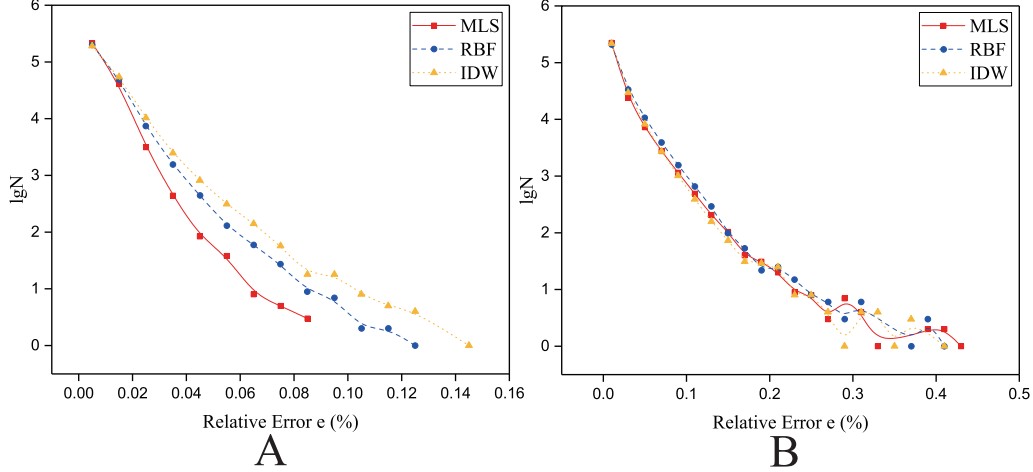

**Figure 14  Frequency distribution of the Relative Error for the parallel implementation developed on a single GPU in the Flat zone.** (A) Regularly distributed and (B) irregularly distributed. The size of data points: Size 1.

density and distribution of the data points, the interpolation accuracy order from high to low is: the MLS interpolation algorithm, RBF, and Shepard's interpolation method.

To further verify the above conclusions obtained from NRMSE, we investigated the relative error of the interpolated results for the same set of data points and interpolation points (i.e., Size 1). The algorithm with the highest accuracy (i.e., the minimum NRMSE) is used to represent the kind of algorithm.

As shown in Figs. 14 and 15, the Y axis is the $lgN$ (N is the count of relative error), and the X axis is the relative error $e$. The $e$ is defined in Eq. (22).

$$e_i = \left| \frac{f_n - f_a}{f_a} \right| \times 100\% \tag{22}$$

where $f_a$ is the theoretically exact solution of the ith interpolated point (the elevation of the DEM at this point), $f_n$ is the predicted value of the ith interpolated point, and $e_i$ is the relative error of the ith interpolated point.

As listed in Tables 8 and 9. For better evaluation of relative error, we also calculated the mean relative error $E$. The $E$ is defined in Eq. (23)

$$E = \frac{\sum_{i=1}^{N_i} e_i}{N_i} \tag{23}$$

where $N_i$ is the number of interpolated points.

### In the flat zone

As shown in Fig. 14, for the flat region, when the data points are evenly distributed, the frequency statistical curve of the MLS is the highest when it is close to zero, the lowest when it is far away from zero, and the relative error distribution range is smaller, which means that the error of MLS method is small. The characteristics of the frequency statistical curve of Shepard's method are completely opposite to those of MLS, which means that the

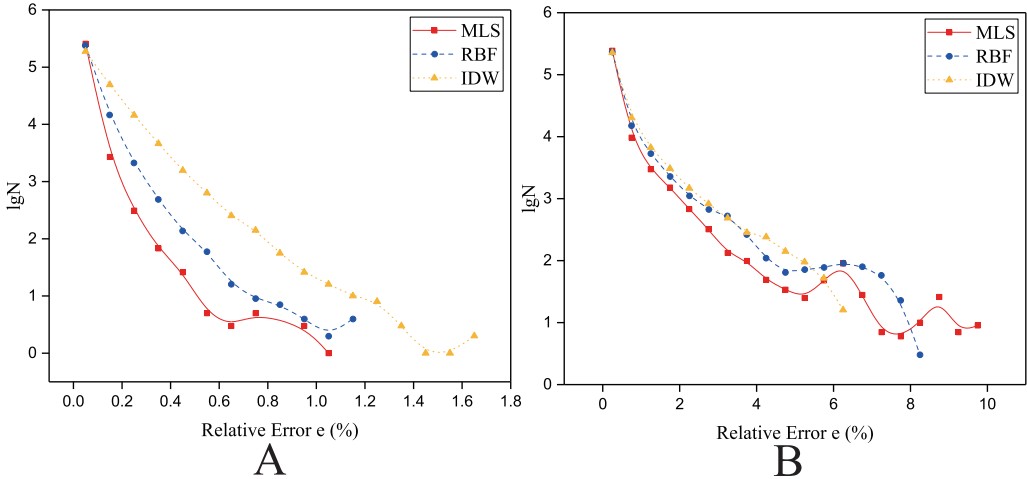

**Figure 15** **Frequency distribution of the Relative Error for the parallel implementation developed on a single GPU in the Rugged zone.** (A) Regularly distributed and (B) irregularly distributed. The size of data points: Size 1.

**Table 8** **The algorithm with the highest accuracy in congeneric algorithms and its corresponding mean relative error in the Flat zone.**

| Distribution | Mean Relative Error E (%) | | |
|---|---|---|---|
| Regularly-distributed | Original MLS | kNNRBF | kNNShepard b |
| | 0.0069 | 0.0078 | 0.0084 |
| Irregularly-distributed | Lancaster's MLS | kNNRBF | kNNAIDW |
| | 0.0144 | 0.0162 | 0.0148 |

**Table 9** **The algorithm with the highest accuracy in congeneric algorithms and its corresponding mean relative error in the Rugged zone.**

| Distribution | Mean Relative Error E (%) | | |
|---|---|---|---|
| Regularly-distributed | Original MLS | kNNRBF | kNNAIDW |
| | 0.0514 | 0.0582 | 0.0904 |
| Irregularly-distributed | Lancaster's MLS | kNNRBF | kNNAIDW |
| | 0.3078 | 0.3493 | 0.3703 |

error of MLS method is large. For the RBF interpolation algorithm, the characteristic of the frequency statistics curve is a transitional phase between those for the MLS and those for Shepard's method. The above curve features and $E$ (Table 8) illustrate that the interpolation accuracy is from high to low in this condition: the MLS interpolation algorithm, RBF, and Shepard's interpolation method.

When the data points are normally distributed, the relative error distribution ranges of all the interpolation methods are larger than that for the uniformly distributed data points. As shown in Fig. 14, the characteristics of the frequency statistics curve of RBF are obvious, the frequency statistical curve of RBF is above the frequency statistical curves of MLS and

Shepard's method, which means that the error of RBF method is larger. The characteristics of frequency statistical curves of MLS and Shepard's method are very similar, and the relative error distribution range of MLS is the largest. As listed in Table 8, in the flat zone, the accuracy of MLS is slightly higher than Shepard's method when the data points are normally distributed.

### In the rugged zone

As shown in Fig. 15, for the rugged region, regardless of whether the data points are uniformly distributed or normally distributed, the characteristics of frequency statistical curves of MLS, RBF and Shepard's method are similar to those illustrated in Fig. 14. However, in Fig. 15B, it is a little different in that most of the frequency statistical curve of Shepard's method is higher than the RBF's. As listed in Table 9, the interpolation accuracy is from high to low: the MLS interpolation algorithm, RBF, and Shepard's interpolation method.

According to the above Figures and Tables, some summary conclusions are obtained as follows:

For the same region, when the density of data points is almost the same, the interpolation accuracy when the data points are evenly distributed is higher than the interpolation accuracy when the data points are nonuniformly distributed.

As listed in Tables 5 and 7, when the data points are evenly distributed, the gap of the accuracy between the three variations of the MLS method, RBF, and Shepard's interpolation methods increases with the decrease of point density.

As shown in Figs. 14 and 15, when the data points are nonuniformly distributed, the maximum relative errors of MLS is larger than other algorithms', however, MLS method has lower NRMSE and E. A small number of larger relative errors has little effect on the overall interpolation accuracy. A large number of small and medium relative errors are the key to determine the interpolation accuracy of the algorithm.

As listed in Table 5, compared with the uniform distribution, when the points are nonuniformly distributed the difference in the accuracy of the interpolation algorithms is not as sensitive to the changes of point density.

Compared with the three variations of the MLS method and the RBF method, Shepard's interpolation method is quite suitable for cases where the data points have a smooth trend. When interpolating for the data points with an undulating trend, the accuracy of Shepard's interpolation method will be poor. When the density of data points is small, this rule becomes more obvious.

## Comparison of computational efficiency

The parallel implementations developed on multi-GPUs is the most efficient, therefore, the parallel implementations developed on multi-GPUs are discussed below.

### In the flat zone

As illustrated in Fig. 12, for the flat region, except for the $k$NNRBF, when the number of data points is not much different, the nonuniformly distributed data point set requires

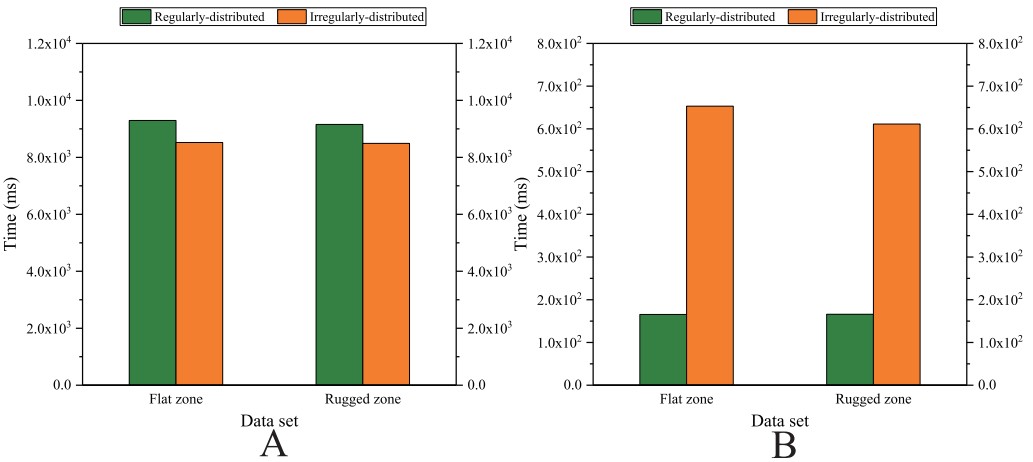

**Figure 16** **Comparison of the running time cost in the *k*NN search procedure.** (A) Sequential version on single CPU and (B) Parallel version on single GPU.

significantly more interpolation time than the uniformly distributed data point set, and with the increase of the number of points, interpolation time does increase as well.

As illustrated in Fig. 10, the speedups achieved by the RBF interpolation method is generally small, and its speedups are not much different in various cases. However, when the size of data point set is Size 1 and the data point set is nonuniformly distributed, the speedup of the RBF interpolation method is larger than other methods, which means that the benefits of parallelism are lower in this case.

As indicated above, the distribution pattern of data points strongly influences the interpolation efficiency.

### *In the rugged zone*

As illustrated in Figs. 11 and 13, the running time and the speedups in the rugged region are almost the same as those in the flat region. In other words, the characteristics of the terrain elevation of data points have a weak influence on computational efficiency.

## Influence of *k*NN search on computational efficiency

According to 'Methods', in the interpolation procedure, the *k*NN search may affect the entire computational efficiency of interpolation.

To specifically evaluate the influence of the *k*NN search on the computational efficiency of the entire interpolation procedure, we investigated the computational cost of the *k*NN search for relatively large numbers of data points, i.e., for the dataset of Size 5 (listed in Fig. 16).

Note that we employ four sets of data points with Size 5, including (1) the set of uniformly distributed data points and the set of nonuniformly distributed data points in the flat region and (2) the set of uniformly distributed data points and the set of nonuniformly distributed data points in the rugged region.

As listed in Table 10, for the sequential version, regardless of whether the data points are uniformly distributed or nonuniformly distributed, the *k*NN search costs approximately

**Table 10  Proportion of the *k*NN search time to the running time of the sequential implementations.** The proportion is $\frac{T_{kNN}}{T_{run}} \times 100\%$, where $T_{kNN}$ is the *k*NN search time, and $T_{run}$ is the running time of the corresponding sequential implementations.

| | Data set | Original MLS | Orthogonal MLS | Lancaster's MLS | *k*NN RBF | *k*NN AIDW | *k*NN IDW | *k*NN shepard1 | *k*NN shepard2 |
|---|---|---|---|---|---|---|---|---|---|
| Flat zone | Regularly-distributed | 74.9% | 78.4% | 73.0% | 28.7% | 77.1% | 96.4% | 90.8% | 92.4% |
| | Irregularly-distributed | 72.8% | 76.0% | 70.8% | 26.8% | 75.1% | 95.5% | 89.7% | 90.9% |
| Rugged zone | Regularly-distributed | 73.7% | 77.2% | 72.0% | 28.3% | 76.2% | 95.3% | 89.7% | 91.0% |
| | Irregularly-distributed | 73.0% | 76.2% | 71.2% | 26.9% | 75.3% | 95.6% | 90.0% | 91.1% |

**Table 11  Proportion of the *k*NN search time to the running time of the parallel implementations developed on a single GPU.** The proportion is $\frac{T_{kNN}}{T_{run}} \times 100\%$, where $T_{kNN}$ is the *k*NN search time, and $T_{run}$ is the running time of the corresponding parallel implementations.

| | Data set | Original MLS | Orthogonal MLS | Lancaster's MLS | *k*NN RBF | *k*NN AIDW | *k*NN IDW | *k*NN shepard1 | *k*NN shepard2 |
|---|---|---|---|---|---|---|---|---|---|
| Flat zone | Regularly-distributed | 46.2% | 41.3% | 44.3% | 6.3% | 54.6% | 65.0% | 63.1% | 62.0% |
| | Irregularly-distributed | 67.8% | 66.8% | 68.3% | 23.1% | 69.5% | 71.0% | 70.3% | 70.4% |
| Rugged zone | Regularly-distributed | 45.8% | 41.2% | 44.4% | 6.3% | 54.6% | 65.3% | 62.5% | 63.3% |
| | Irregularly-distributed | 68.7% | 67.4% | 69.0% | 22.0% | 70.5% | 72.3% | 71.4% | 71.7% |

75% of the computational time of the entire interpolation procedure for the three variations of the MLS interpolation algorithm and the AIDW interpolation algorithm, whereas the *k*NN search costs less than 30% of the computational time for the RBF interpolation algorithm and approximately 90% in the other three variations of Shepard's method. It should also be noted that for the same size of data points, whether they are uniformly or nonuniformly distributed, there is no significant difference in the computational cost of the *k*NN search; that is, the distribution pattern of data points is of weak influence on the computational efficiency of the *k*NN search in the sequential version.

As listed in Table 11, for the parallel version developed on a single GPU, when the sizes of data points are almost the same, it would cost much more time in the *k*NN search when the data points are nonuniformly distributed than when the data points are uniformly distributed. Moreover, when the data points are nonuniformly distributed, the proportion of the *k*NN search time to the total time is approximately 10% to 20% more than the proportion when the data points are uniformly distributed under the same conditions.

On the GPU, for the same interpolation method and the same data size, the proportion of the *k*NN search time relative to the total time when the data points are nonuniformly distributed is larger than that when the data points are uniformly distributed, and the achieved speedups are small.

However, on the CPU, the proportion of *k*NN search time when the data points are nonuniformly distributed relative to the total time is similar to that when the data points are uniformly distributed, and the achieved speedups are similar. This is because there are a large number of logical operations, such as switches in the *k*NN search, and the GPU is inherently not as suitable for performing logical operations as the CPU.

In the *k*NN search procedure, the number of points in the search range is slightly smaller than *k* after determining a certain level. After the level is expanded, the number

of points in the search range will be more than $k$. In this case, the $k$ nearest neighbors should be selected and the redundant neighbors should be ignored by first sorting and then discarding. Unfortunately, there are a large number of logical operations in sorting.

In this procedure of sorting and discarding, when the point density is intensive in a region, the number of found nearest neighbors would be far more than the expected $k$, and much computational time would thus be required to sort the found neighbors.

For areas with sparse data points, it takes more time to find enough $k$ points by expanding the region level. Therefore, in contrast to a uniform distribution, when the data point set is nonuniformly distributed, the $k$NN search needs more computational time and its proportion of the total time is also greater.

## CONCLUSION

In this paper, we present the development of the sequential version, the parallel version on a multicore CPU, the parallel version on a many-core GPU, and the parallel version on multi-GPUs for each of the eight variations of the MLS, RBF, and Shepard's interpolation algorithms. We also evaluated the interpolation accuracy and computational efficiency for the above four versions of each variation when building large-scale DEMs. We have obtained the following observations.

(1) The distribution pattern of data points and the landscape conditions strongly influences the interpolation accuracy. The distribution pattern of data points strongly influences the interpolation efficiency, and the landscape conditions have a weak influence on the interpolation efficiency.

(2) For the same flat region, when the density of points is large, there is no obvious difference in terms of the interpolation accuracy for all interpolation methods. When the data points are uniformly distributed and the density of points is small, the order of the interpolation accuracy from high to low is: the MLS interpolation algorithm, RBF, and Shepard's interpolation method. When the data points are nonuniformly distributed and the density of points is small, the order of the interpolation accuracy from high to low is: the MLS interpolation algorithm, Shepard's interpolation method, and RBF.

(3) For the same rugged region, regardless of the density and distribution of the data points, the interpolation accuracy order from high to low is: the MLS interpolation algorithm, RBF, and Shepard's interpolation method. When the data points are uniformly distributed, the above rules are more obvious than those when data points are nonuniformly distributed.

(4) The Shepard's interpolation method is only suitable for application in cases where the data points have smooth trends. When the data points have uniformly rugged trends, the accuracy of Shepard's interpolation method is rather unsatisfactory, especially in the case when the density of data points is small.

(5) For the same set of data points and interpolation points, the order of computational expense from high to low is: the RBF interpolation method, the MLS algorithm, and Shepard's method Moreover, for the same size of data points and interpolation points, the computational efficiency in the case when the data points are nonuniformly distributed is worse than when the data points are uniformly distributed.

(6) For the same interpolation method, the impact of $k$NN search on the computational efficiency of the CPU versions and the GPU versions is different. Specifically, the percentage of the computational time of $k$NN search relative to the computational time of the entire interpolation procedure in the CPU versions is much smaller than in the GPU versions.

## ACKNOWLEDGEMENTS

The authors would like to thank the editor and reviewers for their contributions to the paper.

### Funding

This research was supported by the Natural Science Foundation of China (Grant Numbers 11602235 and 41772326), and the Fundamental Research Funds for the Central Universities (Grant Numbers 2652018097, 2652018107, and 2652018109). The funders had no role in study design, data collection and analysis, decision to publish, or preparation of the manuscript.

### Grant Disclosures

The following grant information was disclosed by the authors:
Natural Science Foundation of China: 11602235, 41772326.
Fundamental Research Funds for the Central Universities: 2652018097, 2652018107, 2652018109.

### Competing Interests

Gang Mei is an Academic Editor for PeerJ Computer Science.

### Author Contributions

- Jingzhi Tu conceived and designed the experiments, performed the experiments, performed the computation work, prepared figures and/or tables, and approved the final draft.
- Guoxiang Yang analyzed the data, prepared figures and/or tables, authored or reviewed drafts of the paper, and approved the final draft.
- Pian Qi analyzed the data, prepared figures and/or tables, and approved the final draft.
- Zengyu Ding performed the experiments, performed the computation work, prepared figures and/or tables, and approved the final draft.
- Gang Mei conceived and designed the experiments, analyzed the data, prepared figures and/or tables, authored or reviewed drafts of the paper, and approved the final draft.

### Data Availability

The raw measurements are available in the Supplementary Files.

## Supplemental Information

Supplemental information for this article can be found online at http://dx.doi.org/10.7717/peerj-cs.263#supplemental-information.

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
