# Peer review of "Comparative investigation of parallel spatial interpolation algorithms for building large-scale digital elevation models"

_PeerJ Computer Science, doi:10.7717/peerj-cs.263_

## Round 0.1 · original submission · Major Revisions

Both reviewers highlight that the error evaluation methods need to be better described and the reasoning for the kNN search - apart from the issues with the references/literature review.

Furthermore, there seems to be a mismatch in the focus as reviewer 1 outlines: Which is the most important aspect of the research: (i) comparing various methods of parallel computing, or (ii) comparing accuracies among differing landscape characteristics?

As a result of the paper focus decission the discussion should change accordingly. Given the current presentation of methods, results and discussion reviewer 1 is unconvinced of the authors’ conclusions that terrain characteristics and sampling regularity don’t really matter (for either accuracy or computing time).

Both reviewers also suggest to employ statistical tests to identify if dependencies are significant or not. If they can't be employed, please explain why.

And, it seems to me that recent publication mentioned by Reviewer 2 will be helpful for the discussion too: Ghandehari M, Buttenfield BP, and Farmer CJQ (2019) Ghandehari M, Buttenfield BP, and Farmer CJQ (2019) Comparing the Accuracy of Interpolated Terrain Elevations Across Spatial Resolution. International Journal of Remote Sensing 40, 13, pp. 5025-5049.

For the many details please refer to the comments of both reviewers, and in particular respond to what Reviewer 1 suggests. Also, please check if the DEM data are attached in a format that cab be opened in either ArcGIS or QGIS or GRASS.

Reviewer 1 ·

Basic reporting

Some grammar problems, especially missing articles (the, a, an). Overall however the English is readable, and the flow of the article is logical, although the background and methods sections are probably too terse and could be expanded.
I’m confused by the references to literature that cite only author initials (e.g., T, C instead of complete last name and initials (for example Smith, J.). Also problematic that some citations include a publication year and others utilize some sort of code (e.g., 018a) This format for some but not all authors challenged a smooth reading flow, as I had to return to the References and in several cases check online to establish who the author(s) of cited literature actually are. I suggest a consistent format for citations that includes complete last names for all authors. Many of the references are not original, for example Shepard’s algorithm (also called Inverse Distance Weighting) was first described by Shepard, Donald (1968). "A two-dimensional interpolation function for irregularly-spaced data". Proceedings of the 1968 ACM National Conference. pp. 517–524. doi:10.1145/800186.810616, not by Mei G and N 2017. Curiously, the authors cite Shepard (and it is D. Shepard, not S. Shepard, as cited in the appendix). Likewise, kriging was not first derived in 2013 by an author named T but by Georges Matheron in 1960 based on a thesis by Donald Krige (see for example Cressie, N. A. C., The origins of kriging, Mathematical Geology, v. 22, pp 239–252, 1990). Radial Basis functions are originally attributed to Michael J. D. Powell (1977). "Restart procedures for the conjugate gradient method". Mathematical Programming. 12 (1): 241–254. doi:10.1007/bf01593790.
Another point about the literature review is that while articles are cited, there is little discussion about them. For example, to say that Gumus (2013) compared different algorithms seems insufficient in an article about comparing algorithms. What were Gumus’ findings? Similarly, what did Chaplot or Aguilar discover about the associations between landform types and data density on DEM accuracy? Discussing these articles in more detail would really strengthen the introduction and lead the reader toward the authors’ research questions and significance of this current study. The paragraph in lines 65-66 of the introduction seems misplaced – it should introduce discussion about the impacts of terrain type and data density, rather than follow the end of it.

Experimental design

The authors could strengthen the Methods section considerably by expanding on the discussion of the two 30 m DEMs. What agency compiled these DEMs originally? And for readers’ benefit, how do Hebei and Sichuan landscapes differ, given that landscape type is one of the controlling factors in the study? I would also suggest correcting the legend for area a, which appears to have an elevation range of only 30 or so meters – why does the legend extend to 50 meters? The authors state that the elevation range extends to 48 meters, but there are no red pixels in the image, while the legend indicates that elevations above roughly 35 meters would be orange or red. In the second study area, the DEM apparently has been hillshaded, and the lower elevations (blue) appear to be higher than than the upper elevations (red). Removing the hillshade could eliminate the optical illusion, or possibly the legend is inverted? I can’t tell.
I would appreciate more information about the triangular mesh, was this interpolated somehow and if so by what algorithm? Also, what is the spatial distribution of the random samples? Table 4 implies a regular and irregular distribution, but this is not described explicitly. Were samples constrained to limit the number of samples in each 30 m pixel, to avoid over-fitting? Also how were the five sampling densities chosen? Finally, why does the number of interpolated points exceed the number of data points? If the samples were triangulated from the original DEM and then stripped of elevation values, presumably some sort of seeding would be required to anchor the estimations (the interpolated values). But this is also not explained.
The incorporation of the kNN search should also be described in the Methods section, as it figures prominently in the Discussion section. In particular (in the Discussion section) lines 262 -266 could be moved forward to the Methods section and expanded upon. It’s not clear why the kNN search is needed, is my point.

Validity of the findings

What are the units of Table 4? And are these values showing RMSEs? Residuals, or some other metric? If these are meters, then the errors are on the order of hundredths of a millimeter, which seems too small a magnitude (but again, I don’t have a clear sense of the sampling rate, constraints, or which of the several alternatives for each interpolation method were used, so I might have misinterpreted this table).
The authors state that interpolation accuracy for MLS is better than RBF or Shepards (IDW). Whatever the metric, the values for RBF exceed the MLS results. Meaning that either the RBF is more accurate than MLS, or more accurate than IDW. But neither reflects the statement that the progression of accuracy is MLS – RBF – Shepard.
I appreciate seeing the improvement of parallel over sequential computing. It is of course important to balance accuracy against processing speed, especially in situations where modeling inputs are quite large (here, I estimate over 13 million pixels in each DEM). But I wonder if it is necessary to show all of the sequential results, given that one should expect that the HPC computations will complete in a much shorter time. I suggest eliminating Table 5 entirely and simply report the range of average computing times for all ten of the flat and all ten of the rugged experiments. I also wonder about whether the comparison between single and multiple GPUs is really the top priority in the article, whose main purpose I surmised is to compare differing landscape characteristics and data density.
In the discussion section, what is the metric used to assess “relative error”? That is, relative to what? And why show errors for the single GPU, when the multi-GPS solution is faster? Given the article title, I would expect to read a discussion focusing on details of the regular versus irregular sampling in flat and rugged terrain, possibly with a 2-by-2 contingency table, than to see pages of graphs comparing the various parallel solutions.

Additional comments

The authors compare RBF, Moving Least Squares and Shepard’s interpolation algorithms (parallelized) for building DEMs, criteria are terrain type, data density, distribution of errors, processing efficiency. I like the starting premise that specific scene characteristics should be accounted for. However I am not sure that they maintain this as a primary focus throughout their analysis, which seems to get off-track in prioritizing attention to various parallel computing environments and how much each type improves upon sequential computing.
There is some ambivalence among researchers about the term “high” resolution (does it mean large pixels or small pixels?). In this article one assumes it refers to small pixels; but I suggest using the term “fine resolution” versus “coarse resolution” to avoid any confusion among international readers.
It is unclear what the question marks in the Methods section refer to (line 107).
Finding number 1 (line 313 in the Conclusion section) runs counter to numerous articles comparing interpolation of elevation, and this should be noted by the authors. See for example Ghandhari et al 2019 in IJRS. This is not the only article that compares elevation estimation, but it is a recent one and has a good review of recent literature. Finding 3 (line 322-323) claims that the IDW method is only suitable where data points have “smooth trends”. It certainly seems reasonable that IDW will generate higher errors in rugged terrain, but there is uniformly rugged and non-uniformly rugged, and you tested only the former situation. Was a trend surface analysis run on the data to justify this claim?

Reviewer 2 ·

Basic reporting

The background and literature review section is not well written. The citation method is confusing and references are not properly documented because names are abbreviated in the references section. Several references are not cited and overall the paper seems to have been quickly written without much concern to details. Table and figure captions are not complete making them difficult to understand. Additional details about methods must be included.

Experimental design

The experimental design should be improved to include some statistical metric(s) to compare interpolation accuracy of the different interpolation techniques. Methods try to compare the accuracy and efficiency of a several interpolation techniques for elevation data in a flat and a rough section of data.

Validity of the findings

Finding from this work indicate that a regular pattern of sample points produces a more accurate interpolation solution than an irregular pattern, and this result is true for the flat and the rough terrain datasets. Some types of interpolation may be more accurate than others, but statistical testing is not employed to determine significant differences between interpolation techniques. Parallel processing methods using GPUs and CPUs are shown to improve processing speeds when compared to sequential processing methods, but the speedup is not precisely defined in the article.

Additional comments

1. References do not appear to be properly formatted because several citations do not include complete last names for all authors. At least full last names should be included for all authors of each citation. Likewise the reference to each citation in the text cannot have last names abbreviated.
2. Missing several interpolation methods. Missing reference: Lam, N. 1983 article Spatial Interpolation Methods.
3. Missing Shepard 1970 article. Shepard, D.S. 1970. A two-dimensional interpolation function for computer mapping of irregularly spaced data. Papers in Theoretical Geography, Technical report 15, Laboratory for Computer Graphics and Spatial Analysis. Cambridge, Mass., Harvard University.
4. Please expand on line 32. What is meant by spatial structures and types of data. Is this meaning type of remote sensing instrument, data resolution, 3-d point cloud data?
5. Missing reference: Ghandehari M, Buttenfield BP, and Farmer CJQ (2019) Ghandehari M, Buttenfield BP, and Farmer CJQ (2019) Comparing the Accuracy of Interpolated Terrain Elevations Across Spatial Resolution. International Journal of Remote Sensing 40, 13, pp. 5025-5049. https://doi.org/10.1080/01431161.2019.1577581
6. HASM is not defined before use in line 58.
7. Need to expand discussion for single sentence paragraph in lines 65-66.
8. What is L2 - norm in line 78? These symbols are not defined.
9. Upper and lower case for basis function p may not be consistently applied in line 78.
10. Explain what is meant by nodes in line 78. Does this refer to a node in a cluster?
11. Who is Schmidt in line 81? Do we need a reference?
12. In line 78: Consider adding a word, such as “And” to the beginning of “w(x−xk) is the compact-supported weight function.” Otherwise the period looks like an extension of the formula.
13. What does alpha stand for in equations 5 and 6 of line 79?
14. What is f in equation 7 of line 79?
15. Also, it is unclear where ai fits into the Othogonal MLS interpolation algorithm. Please clarify.
16. What is fk in equation 7 on line 79?
17. Missing some spaces in text within line 84.
18. In equations 11 and 12, it appears the square (2) outside the absolute value signs need to be superscripted.
19. Explain what d represents in equation 15.
20. Missing references in line 107
21. Define kNN in line 113, and please explain that method in the paragraph.
22. In line 117, explain the primary contribution found by Mei G (2016). Describe the Structure of Arrays.
23. It is unclear how the randomly sampled points are being used. Are these the values for fa in the NRMSE computations? Please explain further how the samples are used. Are the samples used to generate the triangular mesh?
24. In lines 128-130, why are the coordinate values too large and transformed into a local coordinate system?
25. Line 137 should say “see Table 1”. Currently reads see Table 2.
26. What do the different sizes in Tables 1 and 2 refer to? Sizes go from 1 to 5. Is this just a different number of sample points?
27. Equation (20) in line 160 shows the normalized root-mean-square error (NRMSE), but it is unclear how the theoretically exact value is determined for fa. Please explain how the interpolated values are generated at the locations where the exact values are sampled. I’m guessing that somehow the generated mesh is used to interpolate values at the locations of where the exact values are sampled, but it is unclear how that is being done.
28. In line 170, what do you mean by “not much different”? The scale range is different in Figure 2 for the regularly distributed and the irregularly distributed data. Looks like the regularly distributed method provides better results (lower RMSE) than the irregularly distributed method for both the rugged and flat data.
29. The colors selected in figures 3 and 4 should be adjusted. There are two shades of red, and two shades of blue/purple that are difficult to distinguish between.
30. Interpretation described in lines 189-192 seem incorrect. kNNRBF has faster times than MLS methods in all cases, and MLS methods appear faster than other methods. Lines 190-190 should be revised from “computational time from high to low” to “computational time from fastest to slowest. . . “.
31. Interpretation in lines 197-199 seems incorrect.
32. Line 203, “inaccurate or even incorrect” is two ways of saying the same thing.
33. Figure 4 shows regular and irregular has similar maximum frequency of about 100,000, but the text suggests regular distribution is taller. This interpretation seems incorrect.
34. Interpretation in lines 219-220 is incorrect. Non-uniform distribution generates larger range of errors than uniform distribution as shown in Figure 4.
35. It is unclear what data the authors are interpreting in lines 230-238 where they compare results from different point densities. Is this information that is in the table, because no figures show results from different point densities?
36. Captions for figures 6 and 7 should indicate which CPU or GPU sequential computation method is employed for the displayed results.
37. In figures 10 and 11 and lines 257 to 261, what is meant by speedups? Is this in seconds or percent speedup compared to sequential method?
38. Figure 12: change “Plate zone” to “Flat zone”.
39. Table 9 caption is incomplete. Please include what the proportion is compared to, i.e. sequential implementation compared to what? Likewise please include additional information for Table 10.
40. Line 383 should be the Shepard reference. Author’s improperly abbreviated.
41. Overall, the paper presents some important considerations for evaluating raster data that may be interpolated from a set of points, or which may be the result of a resampling method. Several interpolation methods are compared using different levels of sampling and processing techniques to discern the more accurate and efficient methods. However, the method of accuracy assessment must be clarified to better describe how the truth data are determined and how the interpolation methods are employed. Without a clear definition of the methods it is difficult to understand the results. Furthermore, it seems that some form of statistical method could be used to compare the accuracies of different interpolation approaches and computation methods. Tabulating such a statistical summary could allow the authors to only focus on a few charts that reveal the most important results, rather than charting all solutions, which is difficult to legibly present and puts too much burden on the readers. It appears that the authors visually interpreted results and, in some cases, the interpretations do not appear accurate or are not properly described.

---

## Round 0.2 · Minor Revisions

Both reviewers have highlighted where improvements are needed focusing on different aspects: One points out the need to quantify the description of DEM while the other suggests improvements to the presentation of results, so that conclusions on algorithm choice can be drawn easier. Also both see need for improvements to the introduction section as well as the need for revision of the English.

Reviewer 1 ·

Basic reporting

Some English syntax and grammar problems remain in this version, but many fewer than in the first version. Literature references are much more consistently formatted. Reference and citation list appear to have been edited to correct earlier inconsistencies, and to cite correct original authors, effectively reducing their self-citation problems from the first version. The literature review has also been expanded to provide more informative details about relevant literature. Some tables and figures are not called tables /figures correctly, and one section of pseudocode is called a figure (Figure 1 on page 6).

Experimental design

I suggest justifying the choice of the three tested algorithsms in the Introductory section – why these three and not others? I suggest also spelling out the algorithm names fully in the main text, not just in the abstract, for a smoother flow of text for readers. Additionally, begin each algorithm description in the Background section with a 1-2 sentence description in lay terms of how it works. For example, the Radial Basis Functions operate as a spline, essentially fitting a series of piecewise surfaces to approximate a complex terrain. And so forth. Diving straight into the mathematics might seem impressive, but it’s more helpful to briefly explain in lay terms to those readers trying to understand concepts as well as mechanics.

Thanks for additional information on the source of the DEM data. But the legends in Figure 3 are still confusing in that the same color ramp is applied to 48 m range for one DEM and the 5800 m range in the second DEM. This implies that a very flat landscape is somehow comparable to a very mountainous terrain. And the caption is incorrect, in that both figures are 2.5D (not 3D). I think what the authors are trying to show is one DEM with uniformly high frequency detail, and a second DEM with a mix of low frequency and medium frequency detail (a river network). The inclusion of the hillshade still seems to invert the terrain, giving a visual impression that the river channels are the highest elevations. Or is the legend still inverted? One way to avoid this would be to use a less saturated color ramp, or monochrome ramp.

Validity of the findings

Thanks for giving more detail on the NRMSE metric. And on the regular and irregular sampling in flat and rugged terrain. But while these differences could be readily quantified, the authors have not yet taken that simple step, which would really strengthen the paper.

Line 281 on page 10 should refer to Table 4, not Figure 4. And a question, how do the authors establish that point density is uniformly or non-uniformly distributed? Was a Nearest Neighbor analysis run, or Moran’s I, and if so could those results be described briefly? It’s interesting that interpolation accuracies differ depending on the density, and I expect that terrain uniformity also plays a role , based on my own research, but this isn’t mentioned, even though the two DEMS differ most significantly in uniformity and frequency of detail and the results shown seem to highlight regular and irregular point distribution. So quantitatively speaking, how regular or irregular are the two distributions?

Table 9 compares NRMSE values but here again I don’t see units. And 10 to minus 5th meters (sorry, could not format as an equation on the journal website) would be hundredths of a millimeter, which seems mighty small magnitudes for such large DEMs. The findings need to be clarified as to whether the NRMSEs summarize entire DEMs, or some focal window that is averaged, or… ?

Additional comments

You have responded to many of my comments, and the paper is improved. My major concern with your methods at this point is that you never quantitatively determine regular and irregular point sampling, but could do so easily. Given that this has become a primary highlight of your revised analysis, that quantification becomes warranted.

Reviewer 2 ·

Basic reporting

The English grammar needs some improvements. Improper sentence structure is used in several places. Multiple statements are included in single sentences and these need to be divided into additional sentences.

Definitions are presented well, but need a few suggested adjustments.

Some methods are presented in the discussion section. This should be revised.

Much detail is presented in the results, but too much information is provided. Results need to be more concisely presented. Ranges in y-axis of several side-by-side charts are different and it is not mentioned in the text. It may be best to show these charts with the same y-axis range.

Experimental design

no comment

Validity of the findings

Some refinements are needed in results statements and conclusions.

Additional comments

This paper presents some interesting results, but there is much redundancy in the presentation of the results. Therefore, it is very difficult to read this paper. The introduction should be revised to be more concise. The methods section is fairly well presented. The results section must be reduced to just the most important results. Accuracy should be presented in tables and charts but then only summarized in the text once. The discussion section presents the most important results, but it also describes some methods. I suggest this paper be rewritten to make the result section much more concise. Describe the best solution for each condition once, and indicate how much better the solution is than the other methods. Present processing time of the methods and indicate the speed up of each of the parallel implementations. Overall, the audience will want to know what is the best interpolation method for each terrain type and each point distribution. Processing speed of the interpolation methods is not as important as accuracy estimates when large variations in accuracy are found. It is good to provide an estimate of the needed time for processing, along with how parallel implementations can improve processing speed. However, it seems too much space is used to evaluate speed-up times when the ranges of improved speeds could be easily summarized in a single table or chart. Some suggestions for needed improvements are listed below.

The paragraphs in the introduction are not well organized.

First paragraph redundantly uses ‘fine resolution’. Content in this paragraph could be condensed.

Paragraph 2 indicates a variety of spatial interpolation algorithms exist that behave differently for different data configurations and landscape conditions. Consequently, the accuracy of a DEM is sensitive to the interpolation technique, and it is important to understand how the various algorithms affect a DEM. Therefore, this study is being conducted.

The authors mention two types of interpolation methods: exact and approximate. It may be good to coordinate the introduction to describe and compare the exact methods, and then describe and compare the approximate methods. Subsequently, the test algorithms could be described as exact and approximate. It is unclear if the tested methods are exact or approximate. If exact and approximate distinctions are not important, then they should not be included in the introduction.

Need an ‘and’ at the end of line 111.

Line 184, kNN should be defined in this line, which is the first occurrence of kNN. That is, k-Nearest Neighbor (kNN), and then kNN can just be used in line 195 and subsequent places.

Line 217: need period (.) after were selected. Start new sentence with ‘The topography…’.

Line 226: It is not clear that importing the sample points into the DEM provides the z values that are used for interpolation. I suggest changing line 226 to ‘algorithm in the square region S, and then accessing the corresponding z coordinates from the DEM.’

Likewise, to improve clarity, line 239 should be changed to ‘point is obtained by accessing the z value of the DEM at the associated x and y coordinates.’ I expect the DEM z-values at the mesh points are used as control for testing the accuracy of the interpolated z values. If this is true, then it should be clearly stated as the last sentence of the paragraph.

In lines 240-243, why are the coordinate values too large and transformed into a local coordinate system?

Results lines 270-293: The range of the y-axis for the regularly distributed interpolation points in figures 4 and 5 is less than 1/3 of the range of the y-axis for the irregularly distributed interpolation points. Thus, the regular distribution provides a more accurate solution for both the rugged and the flat areas. This is not pointed out in the results discussion. This needs to be pointed out. Possibly, the charts could use the same range on the y-axis to better accentuate this fact.

Line 302: should be ‘computational time from slowest to fastest is: …’

It would be easier to interpret the speedups if Tables 6, 7, and 8 were presented in charts.

Line 310: need a period (.) after (lower speedup).

Sentence starting in line 342 is not relevant. Original MLS has best accuracy in the rugged zone with irregularly distributed interpolation points.

Line 400, should start ‘A small number of larger’. Remove “It that”. Also put period after ‘interpolation accuracy,” and start new sentence.

Lines 414-417. Statemen is true, except for the kNNRBF, which seems to take the same amount of time for regular and irregular distributed points.

Processing time is important. General estimate of speedup time is important, but would not be major concern in deciding processing method. So figures 9, 10, and 11 are not important.

---

## Round 0.3 · Minor Revisions

Content-wise your article is fine to me now - and almost ready for production. I just see need for 2-3 small changes (its on the English) and I would like a change (increase of size?) and source information on the new Figure 3. I hope you can see my text comments in the attached PDF.

---

## Round 0.4 · accepted · Accept

Thank your for your last changes, and also for the screenshot from ArcGIS, showing how you generated this nice figure. The paper looks good to me now.